# Interferometric excitation fluorescence lifetime imaging microscopy

**Pavel Malý** [1] ✉, **Dita Strachotová**[1], **Aleš Holoubek** [2] **& Petr Heřman**[1]

Fluorescence lifetime imaging microscopy (FLIM) is a well-established technique with numerous imaging applications. Yet, one of the limitations of FLIM is that it only provides information about the emitting state. Here, we present an extension of FLIM by interferometric measurement of fluorescence excitation spectra. Interferometric Excitation Fluorescence Lifetime Imaging Microscopy (ixFLIM) reports on the correlation of the excitation spectra and emission lifetime, providing the correlation between the ground-state absorption and excited-state emission. As such, it extends the applicability of FLIM and removes some of its limitations. We introduce ixFLIM on progressively more complex systems, directly compare it to standard FLIM, and apply it to quantitative resonance energy transfer imaging from a single measurement.

Fluorescence lifetime imaging microscopy (FLIM) is a sensitive imaging technique widely used in life sciences[1–6]. Usually independent of the fluorophore concentration, it allows in-vivo cellular imaging based on intrinsic fluorescence[7] or extrinsic fluorophores[8], probing local environment (such as viscosity, polarity, ion or metabolite concentrations, oxygen imaging, etc.)[9,10], and various intracellular interactions (protein, lipid, DNA, etc.)[11,12]. A powerful application of FLIM is its use for the detection of Förster resonance energy transfer (FRET) by the shortening of the excited state lifetime of the donor fluorophore due to the excitation transfer to the acceptor[1,13,14]. Despite its popularity, FLIM has its limitations. An obvious one is the presence of several emitting species with overlapping emissions and similar lifetimes, which are difficult, if not impossible, to disentangle in a single time dimension. A serious limitation in the FRET application is the case of highly efficient excitation transfer, which shortens the donor lifetime beyond reliable measurement. Furthermore, a common way to prove the FRET presence is to photo-bleach the acceptor and observe the recovery of the longer donor lifetime, with possible photo-conversion of the bleached acceptor presenting a complication[15].

Variants of FLIM have been developed to address these issues. A straightforward one is FLIM with spectral resolution of the emission that can be realized, e.g., by multichannel detection[16,17] or by Fourier spectrometry[18,19]. Another approach tries to separate the species in the data analysis using phasor analysis[20]. While useful for the identification of the emitting species, these approaches still provide information about the emitting state only, and thus most of the substantial FLIM problems remain. To address the absorption, multiplexing techniques have been developed that combine several excitation wavelengths either electronically[21] or interferometrically[22,23]. These, however, only resolve the excitation at a few fixed wavelengths. Meanwhile, in ultrafast nonlinear spectroscopy an approach to probe excitation at all wavelengths at once while keeping the time resolution is known, using double-pulse excitation as in Fourier Transform spectroscopy[24,25]. Recently, such an approach was even applied to single molecules[26,27], with an example combined with time-correlated single-photon counting (TCSPC)[28].

Here, we present an extension of FLIM with broadband interferometric two-pulse excitation, that allows spectrally resolved excitation with all relevant wavelengths present simultaneously within each excitation pulse. Interferometric Excitation Fluorescence Lifetime Imaging Microscopy (ixFLIM) correlates the excitation spectrum with fluorescence decay within a single measurement. At each pixel of the image, it thus provides an additional dimension along which the individual fluorescent species can be resolved. While ixFLIM keeps the advantages of FLIM, the correlation of excitation spectrum and emission lifetime removes some of its limitations. For sensing and imaging applications, the spectra can be monitored to, e.g., recognize the formation of complexes or solvatochromic shifts, and the emitting species can be identified and disentangled by the combination of their lifetime and excitation spectrum. As we demonstrate in detail, ixFLIM

[1]Faculty of Mathematics and Physics, Institute of Physics, Charles University, Prague, Czech Republic. [2]Department of Proteomics, Institute of Hematology and Blood Transfusion, Prague, Czech Republic. ✉e-mail: pavel.maly@mff.cuni.cz

is useful in FRET measurements. In FRET-ixFLIM, the information about transfer efficiency is present in two forms: as a rise of the acceptor signal after donor excitation, and as donor excitation spectrum detected by the acceptor emission. In contrast to the common FLIM, this interlinked information allows the experimenter to measure both highly efficient and highly inefficient energy transfer using the same donor-acceptor pair, as well as to quantify the concentration of the interacting species.

## Results

### Principle of ixFLIM

The principle of the ixFLIM experiment is outlined in Fig. 1. The imaging and detection part is identical to the standard time-domain FLIM, in our case TCSPC detection is coupled to scanning confocal microscope (Fig. 1a). The spatial resolution is diffraction-limited typically to about a fraction of micrometer in the visible, and time resolution can be, depending on the hardware, as low as several picoseconds[29]. As in FLIM, the fluorescence emission is imaged either by a confocal scan or in a widefield mode. The difference between the ixFLIM and standard FLIM lies in the form of the excitation. Both in FLIM and ixFLIM, the excitation needs to be pulsed to resolve the decay in time. However, in ixFLIM the pulses must be broadband to spectrally cover the absorption of all fluorophores of interest (Fig. 1b). The broadband excitation region is separated from the detection spectral region by spectral filters. To resolve the excitation spectrum, ixFLIM uses an approach well known from Fourier transform spectroscopy. The excitation pulse is split into two phase-stable replicas with variable time delay $\tau$ between them, whose spectral interference leads to a periodic spectral modulation of the detected fluorescence at each delay time[28,30] (shaded spectra in Fig. 1b). The excitation spectrum is then obtained by Fourier transform along $\tau$ as follows. For each delay $\tau$ of the excitation pulses, a frame of FLIM image is recorded. The total raw ixFLIM data are thus in form of a four-dimensional dataset $\text{ixFLIM}_{\text{raw}}(x, y, t, \tau)$, where (x, y) are the coordinates of the 2D image plane, $t$ is the photon arrival time (from TCSPC) and $\tau$ the excitation-pulse delay. The details of the signal processing as well as the implementation of the experiment in our laboratory are described in detail in the Methods section and

illustrated in section S1 of the Supplementary Information (SI). The dataset is Fourier-transformed along $\tau$, yielding the four-dimensional hypercube $\text{ixFLIM}_{\text{raw}}(x, y, t, \omega_\tau)$. After conversion from frequency $\omega_\tau$ to wavelength $\lambda_\tau$, the data are divided by the known imprinted laser excitation spectrum along $\lambda_\tau$, yielding the ixFLIM signal $\text{ixFLIM}(x, y, t, \lambda_\tau)$ resolved along the excitation wavelength $\lambda_\tau$ and emission time $t$. At each pixel (coordinates x, y) of the image, the correlation of the excitation spectrum (along $\lambda_\tau$) and fluorescence decay (along $t$) is thus obtained (Fig. 1c). Importantly, the excitation wavelength is not scanned sequentially, but all wavelengths are present in each pulse, with varying amplitude. The ixFLIM dataset contains the standard FLIM as its marginal, $\text{FLIM}(x, y, t) = \int \text{ixFLIM}(x, y, t, \lambda_\tau)d\lambda_\tau$. The marginal along the other dimension is the excitation-spectrum-resolved interferometric-excitation fluorescence image (ixFIM): $\text{ixFIM}(x, y, \lambda_\tau) = \int \text{ixFLIM}(x, y, t, \lambda_\tau)dt$.

### Information in ixFLIM: Oxonol VI binding to albumin

To demonstrate how ixFLIM works, we measured fluorescence dye Oxonol VI (Oxo VI) in the presence and absence of bovine serum albumin (BSA). Oxo VI is a cationic potential-sensitive redistribution probe that penetrates biological membranes and, according to the membrane potential, partitions between cell/organelle exterior and interior where it binds to cellular constituents such as lipids or proteins[31,32]. When bound, its absorption spectrum shifts to the red [Fig. S5 in the SI] and fluorescence lifetime prolongs[33]. The magnitude of these changes is often used empirically to assess the relative fraction of bound vs free fluorophore.

We imaged two adjacent microcapillaries, one filled with free Oxonol VI in a PBS buffer and the other with Oxo VI bound to BSA (Fig. 2a), with absorption of both the bound and free Oxo VI covered by the broadband excitation laser spectrum (full lines in Fig. 2b). The red tail of the fluorescence emission (dashed lines) above 700 nm was detected. The difference between the bound and free oxonol is visible in FLIM, which is obtained from the ixFLIM dataset by integration over $\lambda_\tau$. The mean (first moment) lifetime, $\text{FLIM}(x, y, \langle t \rangle)$, can be used for FLIM imaging without any fitting, distinguishing the bound and unbound oxonol clearly (Fig. 2c). The direct analogy in the spectral

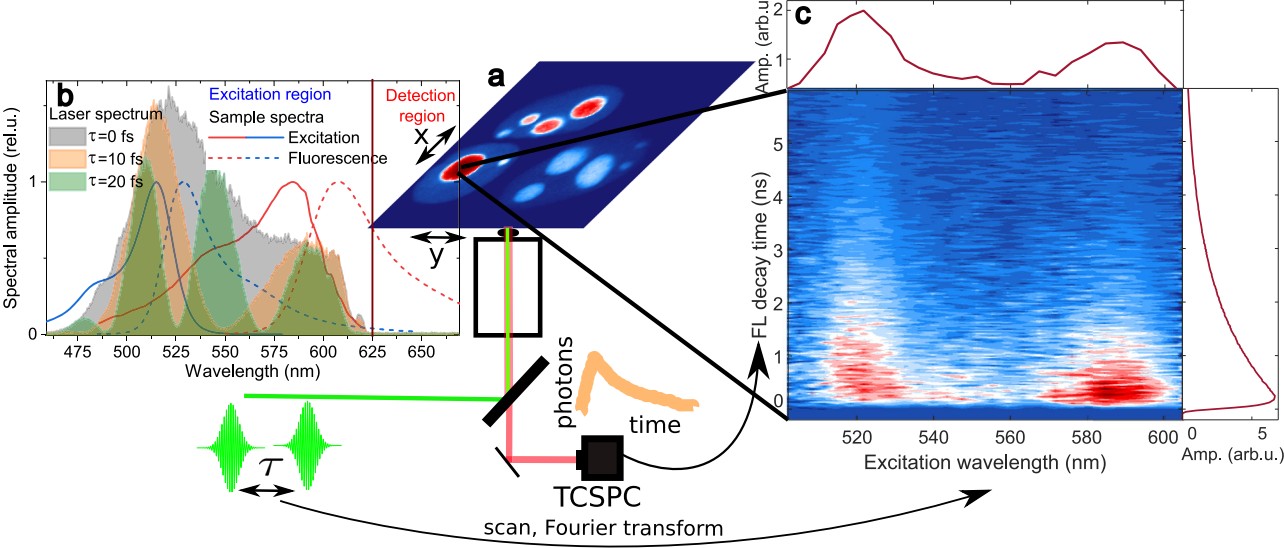

**Fig. 1 | Principle of ixFLIM. a** Standard time-domain FLIM with point scanning (x,y) and TCSPC detection. **b** Spectral modulation of broadband excitation pulses (shaded spectra) by interference of two overlapping pulses delayed by $\tau$. The laser spectrum (shaded regions) must cover the excitation spectrum of the sample (solid lines), and the fluorescence emission (dashed lines) should be separated by a long-pass filter. **c** Fourier transform of the time resolved FLIM data leads to the ixFLIM transient map that correlates the fluorescence excitation spectrum (horizontal) and emission decay (vertical) at each pixel of the image. The marginals of the transient map represent the total fluorescence decay (vertical) and excitation spectrum (horizontal). Exemplary data of HEK-293T cells that are described in detail in the results.

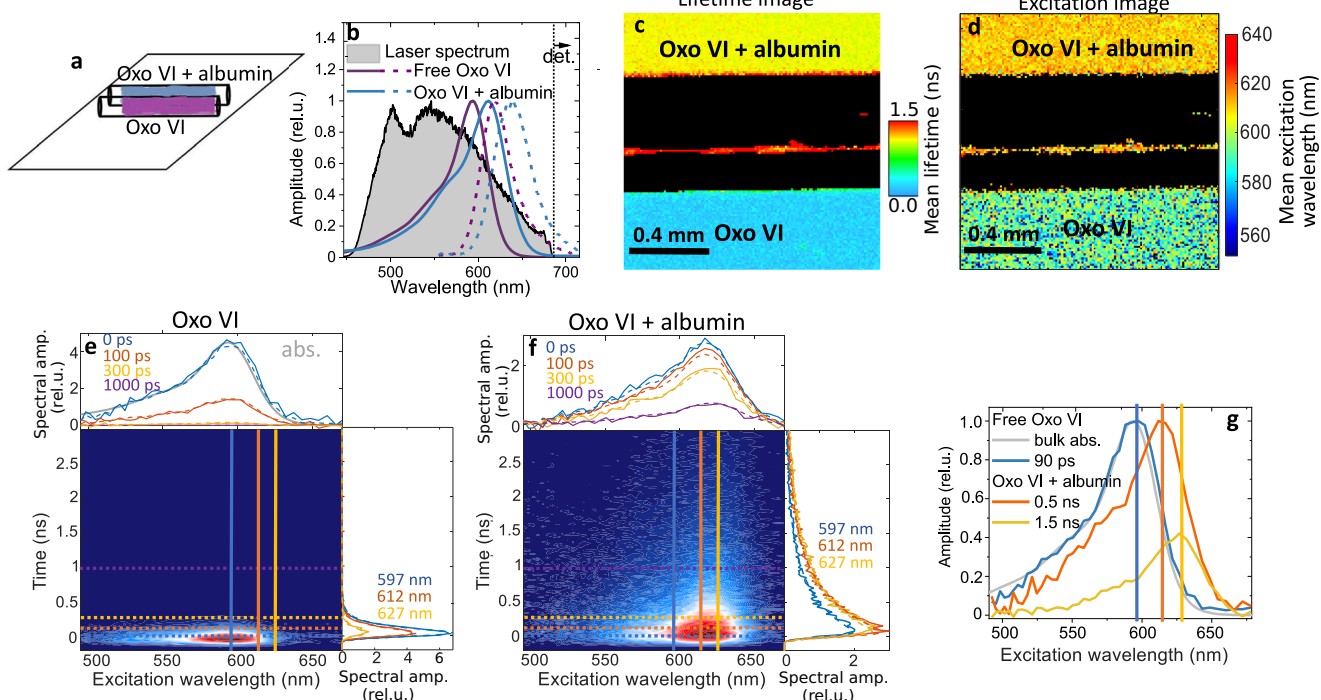

**Fig. 2 | ixFLIM measurement of Oxonol VI. a** sketch of the microcapillaries filled with free (purple) and bound (turquoise) Oxo VI. **b** Bulk absorption (solid lines) and fluorescence (dashed lines) spectra of free (purple) and bound (turquoise) oxonol, covered by the broadband excitation spectrum (black). **c** FLIM image $FLIM(x, y, \langle t \rangle)$ color-coded by mean lifetime. **d** fluorescence excitation spectrum image $ixFIM(x, y, \langle \lambda_\tau \rangle)$ color-coded by mean excitation wavelength. In (**c**), (**d**) black indicates intensity lower than given threshold (5% maximum counts per pixel). **e** Transient map $ixFLIM(t, \lambda_\tau)$ of free Oxo VI (from the bottom capillary), with

spectral and temporal cuts at the marked positions, overlaid by the global analysis fit (dashed lines). **f** Same as in (**e**) but for the upper capillary with bound Oxo VI. **g** Decay-associated spectra of the time components from the global analysis of the free Oxo VI (blue, compared to its absorption spectrum in gray), and Oxo VI bound to BSA (red and yellow). The global analysis has been done on the maps in (**e**) and (**f**), and the position of the corresponding vertical lines marks the maxima of the transient spectral components. The number of experimental repetitions is described in the Statistics and Reproducibility section in Methods.

domain is to image by the mean (center of mass) excitation wavelength, $ixFIM(x, y, \langle \lambda_\tau \rangle)$, obtained from the temporally integrated ixFLIM image, as shown in Fig. 2d. Clearly, the regions of bound and unbound Oxo VI can be distinguished both by the mean fluorescence lifetime and by the mean excitation wavelength. However, there is much more information present in the ixFLIM dataset. For two regions of interest coinciding with the two micro-capillaries, we show the transient excitation maps obtained by data integration over x and y in Fig. 2e (free Oxo VI) and Fig. 2f (bound Oxo VI). Common techniques can be used to analyze the transient map such as global analysis that decomposes the signal into a sum of exponentially decaying components,

$$ixFLIM(t, \lambda_\tau) = \sum_i S_i(\lambda_\tau) e^{-k_i t} \otimes irf(t). \quad (1)$$

Here, $S_i(\lambda_\tau)$ are the spectra associated with the individual components that decay with rates $k_i = \tau_i^{-1}$. The exponential decay is convolved ($\otimes$) with the instrument response function, which in our case can be well described by an approximately 50 ps wide Gaussian. The fits are shown as dashed lines in Fig. 2e, f, and the spectra of the individual components in Fig. 2g. The free Oxo VI is represented by a single species with 90 ps lifetime and absorption peaking around 595 nm (blue in Fig. 2g), in agreement with literature[31,33,34]. In contrast, the Oxo VI bound to BSA is present in two species with lifetimes of 0.5 ns and 1.5 ns, with spectra progressively red-shifted to 620 nm and 632 nm, respectively (red and yellow in Fig. 2g). This correlates perfectly with the two known major binding sites of albumin[35], which apparently lead to a different magnitude of the spectral shift, correlating with lifetime prolongation. The presence of multiple spectra and

kinetics is important when using Oxo VI as a membrane probe. While we used Oxo VI and albumin, the same principle (spectral and lifetime shifts, multiple binding sites) applies to most of other proteins and fluorophores. Beyond merely illustrating the principle of ixFLIM, this example highlights its utility in disentangling different species. Importantly, the same type of transient map is contained in each pixel of the image.

### FRET-ixFLIM molecular ruler: Cy3–Cy5 at DNA

The Oxo VI example served to illustrate the type of information contained in the ixFLIM dataset. We proceed with application to Förster Resonance Energy Transfer (FRET)[36], which is commonly used to measure nanometer-scale distances in biostructures, as well as to evaluate interaction between labeled biopolymers[1,14]. The excitation energy transfer rate $k_T$ between donor and acceptor fluorophores at the distance $R$ apart is

$$k_T = k_D \left( \frac{R_0}{R} \right)^6, \quad (2)$$

where $k_D$ is the fluorescence decay rate of the donor and $R_0$ is a so-called Förster radius, which can be expressed as[3,37]

$$R_0^6 = 8.79 \cdot 10^{-5} \left[ \hat{\mu}_a \hat{\mu}_d - 3 \left( \hat{\mu}_a \cdot \hat{R} \right) \left( \hat{\mu}_d \cdot \hat{R} \right) \right]^2 n^{-4} \int d\lambda \, \alpha_a(\lambda) FL_d(\lambda) \lambda^4. \quad (3)$$

Here, $\alpha_a(\lambda)$ and $FL_d(\lambda)$ are the absorption spectrum of the acceptor, and normalized emission spectrum of the donor, respectively. $\hat{\mu}_{a,d}$ are the acceptor and donor transition dipole moments

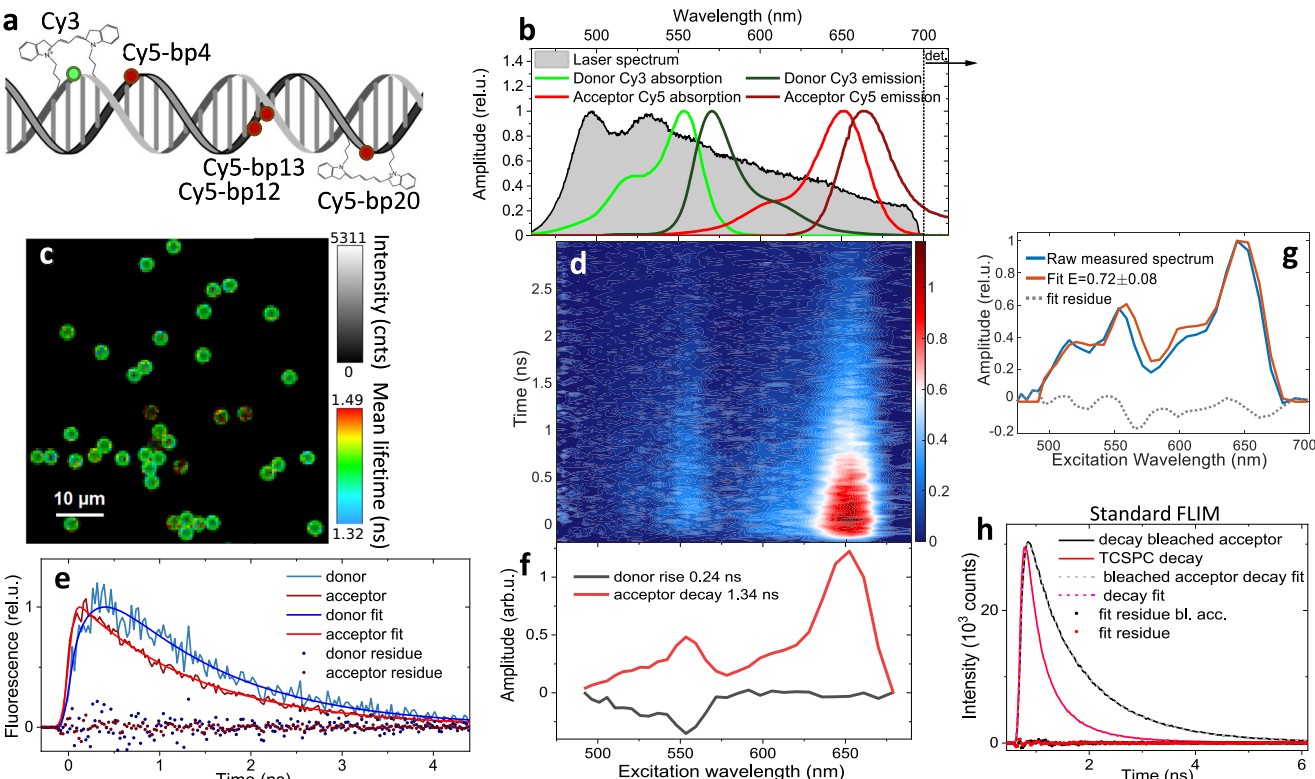

**Fig. 3 | ixFLIM FRET demonstration on internally labelled Cy3–Cy5 DNA constructs. a** Schematic geometry of the dye attachment to the DNA. The numbers indicate the Cy3–Cy5 mutual distance in base pairs (bp). Chemical structure adapted from https://en.wikipedia.org/wiki/File:Cy3_Cy5_dyes.gif by "Nwbeeson". **b** Cy3 (green, shorter wavelength) and Cy5 (red, longer wavelength) absorption (light shade) and emission (dark shade) spectra, overlaid by the broadband excitation pulse spectrum (black), truncated by a 700 nm short-pass filter to facilitate excitation of both donor and acceptor and detection of longer-wavelength Cy5 emission only. **c** FLIM image $FLIM(x, y, \langle t \rangle)$ of the beads to which the DNA constructs are attached with 20 bp Cy3–Cy5 distance. **d** ixFLIM transient map $ixFLIM(t, \lambda_\tau)$ obtained from the image in (**c**). **e** Transients in the donor and acceptor spectral region (as identified spectrally according to (**b**), overlaid with fits from global analysis of the ixFLIM data from (**d**). **f** Spectral components from the global analysis, 240 ps rise of the donor signal due to FRET (black), 1.34 ns decay of the emitting acceptor (red). **g** Spectral fit (orange) to extract E according to Eq. (6), decomposing the raw excitation spectrum (i.e., not divided by the laser spectrum, blue) into the weighted sum of the donor and acceptor absorption spectra multiplied by the laser spectrum. Fit residuals are shown in gray. **h** A complementary standard full-frame FLIM measurement with 530 nm excitation and detection of donor emission. Shown is the donor emission decay in the presence of acceptor (red) and with acceptor bleached (black). The curves are overlaid with fits (dashed) as described in the SI, fit residuals are shown as well (dotted). The number of experimental repetitions is described in the Statistics and Reproducibility section in Methods.

normalized to unity, $\hat{R}$ is the normalized donor-acceptor distance, and $n$ is the refractive index. While the $\frac{1}{R^6}$ dependence in Eq. (2) allows the distance measurement, the orientation factor in the square brackets of Eq. (3) provides sensitivity to the dipole geometry. The transfer efficiency $E$ is given by the competition of the transfer and donor excited state decay:

$$E = \frac{k_T}{k_T + k_D} = \frac{1}{1 + \left(\frac{R_0}{R}\right)^6}. \tag{4}$$

To demonstrate the capability of quantitative FRET-ixFLIM imaging, we apply it to a Cy3–Cy5 donor–acceptor pair, which is commonly used for DNA labeling. The dyes can be attached to the sugar-phosphate backbone as internal labels, at specific positions. The FRET between them then reports on the local DNA conformation[38–43]. In our work, we have designed internally labelled DNA oligonucleotides[44] with the donor Cy3 and acceptor Cy5 covalently linked to the sugar-phosphate backbone of complementary DNA strands at determined positions (see Fig. 3a and Methods). Upon hybridization, these formed a double-stranded DNA with donor-acceptor pairs at distances of 4 base pairs (bp04), 12 bp (bp12), 13 bp (bp13), and 20 bp (bp20). Similar samples with a different dye pair have been used recently in a single-molecule FRET benchmark study[45]. Our broadband excitation

comfortably covers absorption of both the Cy3 and Cy5 (Fig. 3b), and, in contrast to standard donor-lifetime FLIM, we detect emission from the acceptor Cy5 only. For imaging purposes, the DNA constructs were attached by a biotin-streptavidin link to magnetic beads (2.8 μm in diameter) with the size comparable to cellular structures, e.g. nucleoli[13,46]. Such beads can be easily imaged, as shown on the bp20 beads in Fig. 3c. This image, obtained by spectrally integrating the ixFLIM dataset, shows a uniform narrow distribution of lifetimes around 1.4 ns (Fig. S7 in the SI), as expected from the acceptor-detected FLIM[47–49]. The ixFLIM transient map obtained from all the beads in Fig. 3c is shown in Fig. 3d. Two clear absorption bands are visible, corresponding to the donor and acceptor spectra. At longer delay times, the signal decays exclusively with the acceptor lifetime, as expected from the detection spectral region. We have indeed verified that no more than 3% of the emission comes from the Cy3 donor (Fig. S14 in the SI). The presence of the donor Cy3 peak in the ixFLIM spectrum is therefore immediately a qualitative proof that the FRET takes place.

There are two ways to quantify the FRET efficiency in the ixFLIM measurement. First, from the signal kinetics. Since the excitation of the donor can lead to acceptor emission (with rate $k_A$) only after the energy is transferred, the donor signal appears with a delay with respect to the acceptor fluorescence. As derived in the Methods (Eq. 10), if the direct emission of the donor is negligible, the FRET-

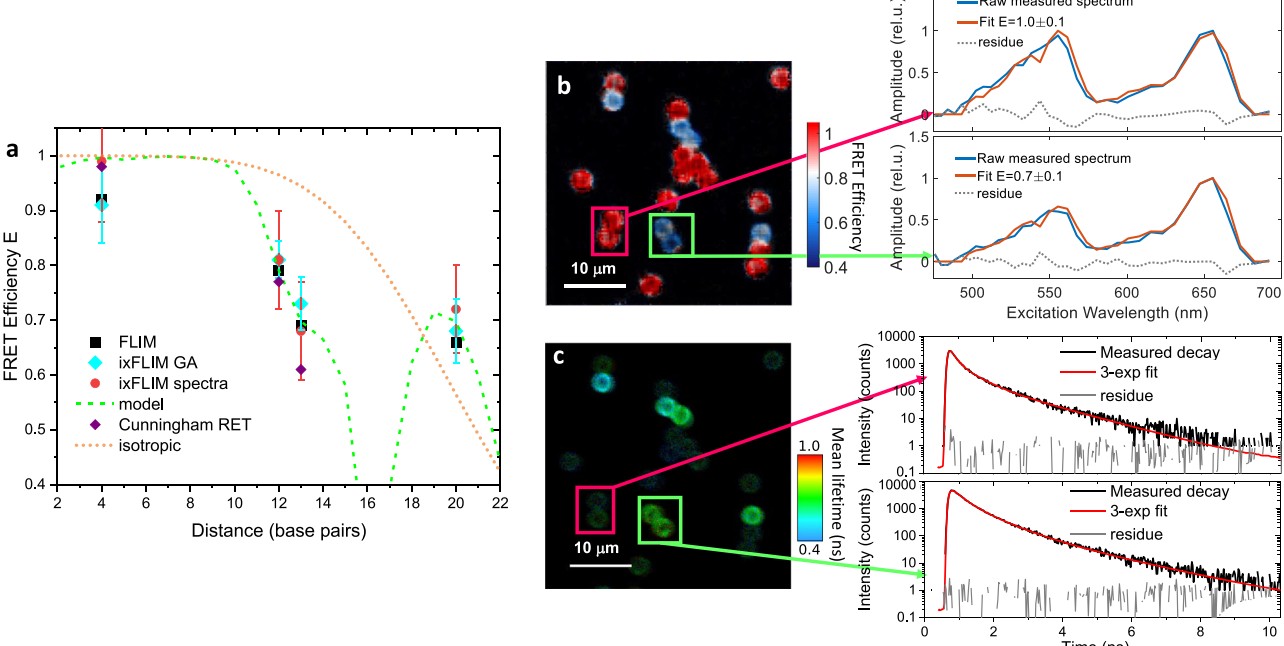

**Fig. 4 | ixFLIM-based FRET imaging. a** FRET efficiency dependence on the donor-acceptor distance along the DNA. Values obtained by ixFLIM global analysis (cyan), ixFLIM spectral fit (red), donor lifetime FLIM (black) and by calculation (green dashed line) are compared between themselves and with values from literature, ref. 38 (purple). To illustrate the sensitivity to geometry, the dependence for a hypothetical isotropic molecular orientation is shown as well (orange). The error bars represent the standard deviation, obtained either directly from the standard error of the fit (ixFLIM spectra, red), or from the fit error of the FRET rates using error propagation (ixFLIM GA and FLIM); for the FLIM data the error bars reach 25%

of the values (see Table 1) and were omitted for clarity. **b** ixFLIM-obtained image of a mixture of beads with Cy3–Cy5 donor–acceptor pairs separated by 4 bp and 20 bp. The image is color-coded by FRET efficiency extracted in each pixel from the ratio of the donor and acceptor excitation spectra. **c** The same sample measured by FLIM, determination of the bp04 lifetime is unreliable due to very low intensity and short lifetime caused by the FRET efficiency close to 1. Two pairs of beads are marked in both images with FRET efficiency of $E = 0.7$ (green, 20 bp) and $E = 1$ (red, 4 bp).

ixFLIM signal detected by acceptor fluorescence can be expressed as

$$\text{ixFLIM}_A(t,\lambda_\tau) = k_A \phi_A \eta_A c_A \left\{ \epsilon_A(\lambda_\tau)e^{-k_A t} + \epsilon_D(\lambda_\tau)\frac{k_T}{k_T + k_D - k_A}\left(e^{-k_A t} - e^{-(k_T + k_D)t}\right) \right\}. \tag{5}$$

Here, $\phi_A$ is the acceptor fluorescence quantum yield and $\eta_A$ detection efficiency, $c_A$ is the acceptor concentrations (in our case of Cy3–Cy5 pairs the donor concentration $c_D = c_A$ by design) and $\epsilon_{A,D}(\lambda_\tau)$ are the excitation spectra of the donor and acceptor, respectively. There are therefore just two time components in the kinetics: one that contains both donor and acceptor spectra and decays with the lifetime of the acceptor, and a second one that rises with the sum of the donor decay rate and the FRET transfer rate[50]. If the donor decay rate is known, e.g., from an independent measurement, the FRET rate $k_T$ can thus be determined. This is demonstrated in Fig. 3e, where the fits of the transients at the peak positions of the donor and acceptor clearly show the delayed rise of the donor peak due to the FRET. The fitting was done on the whole transient map using global analysis [Eq. (1)], in which the two components have been identified (Fig. 3f). The slower component (red line) reflects the decay of the Cy5 acceptor. The faster component (black line) has negative spectral amplitude signifying signal rise and spectrum of the donor (compare to Fig. 3b). This is thus the component connected to the FRET, with the risetime of $\tau_{\text{rise}} = (k_D + k_T)^{-1} = 235 \pm 24$ ps. Considering the known donor lifetime of $k_D^{-1} = 0.75 \pm 0.08$ ns (see the measurement of Cy3-labelled beads only in Fig. S13 in the SI), we obtain FRET efficiency of $E_{\text{ixFLIM-GA}} = 1 - \frac{k_D}{k_T + k_D} = 0.68 \pm 0.07$.

A second way to quantify the FRET efficiency is from the time-integrated excitation spectrum ixFIM$(x, y, \lambda_\tau)$, which for negligible

detected donor emission reads [see Eq. (12) in Methods]:

$$\text{ixFIM}_A(\lambda_\tau) = \int \text{ixFLIM}_A(\lambda_\tau, t)dt = \phi_A \eta_A c_A \{\epsilon_A(\lambda_\tau) + \epsilon_D(\lambda_\tau)E\}. \tag{6}$$

The contribution of the donor relative to the absorption spectrum determines the transfer efficiency $E$. For this, the absorption spectra of the donor and acceptor have to be known, as is the case of commonly-used FRET probes[51]. To accommodate possible spectral shifts upon DNA incorporation, we have measured the excitation spectra independently on samples with non-hybridized oligonucleotides with either Cy3 or Cy5 dye only (Fig. S8 in the SI). The example of the fit of the excitation spectra obtained from the time-integrated transient map is given in Fig. 3g, getting the efficiency of $E_{\text{ixFLIM-Spec}} = 0.72 \pm 0.08$. Note, that the two approaches are to a degree complementary. Fast, efficient transfer is difficult to discern in the time domain, but the donor spectrum will be present strongly so that the spectral fit is reliable. On the other hand, slow inefficient transport can be conveniently tracked in time domain, where spectral integration over the donor peak will increase the signal to noise ratio. In between, there is a wide region of transfer efficiencies where both approaches work well, as in our case of 20 bp distant dyes. In this region, the combination of the two approaches can be used to eliminate an additional unknown parameter, such as unknown change in spectrum or lifetime. In this sense, the FRET-ixFLIM has a strong intrinsic reference that makes it more robust than standard FLIM-FRET.

To further verify the ixFLIM approach, we have measured the beads using standard donor lifetime FLIM as well (Fig. 3h). In this measurement, we excited the beads with 530 nm narrowband (20 nm) pulses and detected emission in the 560 nm – 600 nm spectral

**Table 1 | FRET efficiency as a function of Cy3–Cy5 position along the DNA: summary of results**

| Cy3–Cy5 distance | ixFLIM Signal rise fit | ixFLIM Spectral fit | FLIM | Theory |
|---|---|---|---|---|
| 4 bp (1.36 nm) | 0.91 ± 0.07 | 0.99 ± 0.11 | 0.92 ± 0.27 | 1.0 |
| 12 bp (4.08 nm) | 0.81 ± 0.03 | 0.81 ± 0.09 | 0.79 ± 0.41 | 0.78 |
| 13 bp (4.42 nm) | 0.73 ± 0.05 | 0.68 ± 0.09 | 0.69 ± 0.26 | 0.69 |
| 20 bp (6.8 nm) | 0.68 ± 0.06 | 0.72 ± 0.09 | 0.66 ± 0.19 | 0.71 |

window, where the acceptor emission is absent. An issue was an appreciable auto-fluorescence of the beads themselves, which was significant under a standard 488 nm excitation. The situation improved with the 530 nm excitation, but still necessitated subtraction of the empty bead background (see the SI for details). After acquiring the FLIM data, we fully bleached the acceptor by a strong 560 nm light (as can be verified by ixFLIM, see Fig. S14 in the SI). Repeated FLIM measurement led to a prolonged mean lifetime arising from donor-only emission. As described in the SI, from the donor lifetime shortening we inferred the FRET efficiency. For the bp20 sample, we obtained $E_{FLIM} = 0.66 \pm 0.19$, which is in a good agreement with the ixFLIM values. Note, however, the large error of the recovered value, stemming from the unreliable decomposition of the one-dimensional decay into three decay components needed for the fit. As a yet another check, we measured a sample where the donor and acceptor were attached to adjacent beads separated by the large 0.5 μm distance which made FRET practically impossible. As shown in Fig. S17 in the SI, in the absence of FRET the transport-associated donor rise component is not present in the ixFLIM map.

So far, we have discussed the oligonucleotides with distance between the donor and acceptor fixed to 20 bp. With the methodology for determination of the FRET efficiency from the ixFLIM (and FLIM) data established, we varied the donor-acceptor distance along the DNA in order to study FRET efficiency as a function of distance and geometry, similar to Refs. [38,45,52]. The donor (Cy3) position was fixed, and the acceptor separation decreased from 20 bp to 13 bp, 12 bp and 4 bp. (Fig. 3a and Methods). For all four positions, we extracted the FRET efficiency by ixFLIM from the spectral fit and from the signal rise using global analysis as before, see section S4 of the SI. We also measured the efficiency by the standard donor-FLIM for verification. The dependence of the FRET efficiency obtained by all three ways is shown in Fig. 4a. Clearly, there is a good agreement between the inferred values, validating the ixFLIM approach. From the Förster theory, the dependence of the transfer efficiency on the dye position can be calculated. The detailed work by Cunnigham et al.[38] has shown that for separations larger than 6 base pairs the Förster theory applies, and the average Förster radius is $R_0 = 7.1$ nm, which is significantly larger than in solution. This is explained by the longer excitation lifetime due to the increased rigidity of the covalently attached dye. The measured values do not follow the curve that assumes isotropic random dye orientation (orange in Fig. 4a). This is expected since the internal labeling, where the rigid attachment of the dyes to the helix backbone defines both the fluorophore separation and their mutual orientation, which changes along the helix. On the other hand, due to the inevitable structural disorder, there is a distribution of the dye orientations, which is a priori unknown[38,52]. In the calculation, the DNA geometry was fixed to that of B-type DNA and the orientational disorder and the precise distance of the fluorophore transition dipole from the DNA axis were taken as fit parameters. These were optimized by a global least-square algorithm to fit experimental data (see section S4 of the SI). The results are compared to the experimental values in Fig. 4a. All results are summarized in Table 1.

Clearly, the agreement between the experiment and theory is very good, verifying the quantitative performance of ixFLIM in FRET. For the 4 bp dye separation the Förster theory with dipole-dipole coupling might not hold anymore. There is, however, no acceptor fluorescence quenching at this distance and near-100% efficient transport, same as expected from Förster theory. In Fig. 4a, we further compare the data to those measured by Cunningham et al.[38], showing a good agreement, with some difference possibly stemming from the donor and not acceptor position being changed in their work. The recovered parameters of our model, 30° disorder in the helical angle and negligible disorder in the rise angle, provide a useful foundation for DNA studies using such labeling.[39,42,43,53].

For the accurate determination of the FRET efficiency, we used data averaged over 10-30 beads (Fig. 3b for 20 bp and Fig. S7 in the SI for 4, 12 and 13 bp). However, the same information, albeit noisier, is present in each pixel of the image. One can thus use the spectral method to determine $E$ and use it for coloring the image by FRET efficiency. To demonstrate this, we prepared a mixture of bp04 and bp20 beads. In Fig. 4, we compare imaging of this mixture by ixFLIM (Fig. 4b), and by standard donor FLIM (Fig. 4c). Since the FLIM three-component one-dimensional fit cannot be performed reliably on single-pixel decays, the FLIM reports on the mean lifetime variations only. In contrast, in ixFLIM we can simply integrate the donor and acceptor spectral regions in the temporally integrated ixFLIM dataset. The resulting image clearly distinguishes the two types of beads. Note, that for the most efficient FRET (nearly 100%), the donor-based FLIM is very unreliable because of the strong FRET-induced emission quenching. In contrast, ixFLIM can easily observe FRET in constructs with both fast and slow transfer.

## Protein interaction: nucleophosmin in nuclei and nucleoli of HEK-293T cells

In many biological and biophysical applications, FRET is used for detection of protein interactions and formation of macromolecular assemblies[11,14]. To demonstrate the applicability and advantage of ixFLIM in such situations, we imaged HEK-293T cells co-transfected with the abundant nucleolar protein nucleophosmin (NPM), where NPM monomers are tagged with donor mVenus or the acceptor mRFP1 (see Methods for sample preparation). With this dual labeling, one can study the oligomerization state of NPM in cell nuclei and nucleoli[13]. It is assumed that NPM forms higher oligomers, mainly pentamers or decamers in nuclei[54]. If that is the case, there should be a significant fraction of the mixed oligomers with donor-acceptor pairs and thus FRET should be present[13,55]. Compared to the Cy3–Cy5 pairs, the situation is somewhat more complicated for the mVenus-mRFP1 pair. The donor mVenus has longer fluorescence lifetime (2.9 ns)[56] and higher quantum yield (0.64)[51] than the lifetime (1.5 to 1.8 ns)[57] and quantum yield (0.25)[51] of the acceptor. Their smaller spectral separation results in both donor and acceptor emission being present in the collected data. Furthermore, while in the DNA constructs we had always a donor-acceptor pair present, here we have donor and acceptor in different stoichiometries randomly present in the individual NPM oligomers, including some fractions of non-interacting donors and non-interacting acceptors. Finally, NPM oligomers contain non-labeled endogenous NPM as well. This real-world application is thus a good test for ixFLIM.

We have measured the same HEK cells both by ixFLIM and then by FLIM, which is possible due to the noninvasive low-intensity nature of these measurements. The exemplary HEK cell nucleus with a roundish nucleolus is shown in Fig. 5a measured by FLIM and in Fig. 5b by ixFLIM. For the ixFLIM we excited both the donor and acceptor (Fig. 5c) and detected the emission using a 655 nm longpass filter. In the FLIM measurement, we excited by the standard 488 nm pulsed excitation, and detected in the donor mVenus channel (500 nm to 560 nm). The FLIM image (Fig. 5a) shows the mean lifetime around 2.7 ns (decay in Fig. 5d, red), slightly shorter in the nucleolus. In ixFLIM, the mean excitation wavelength is about 555 nm (spectrum in Fig. 5e,

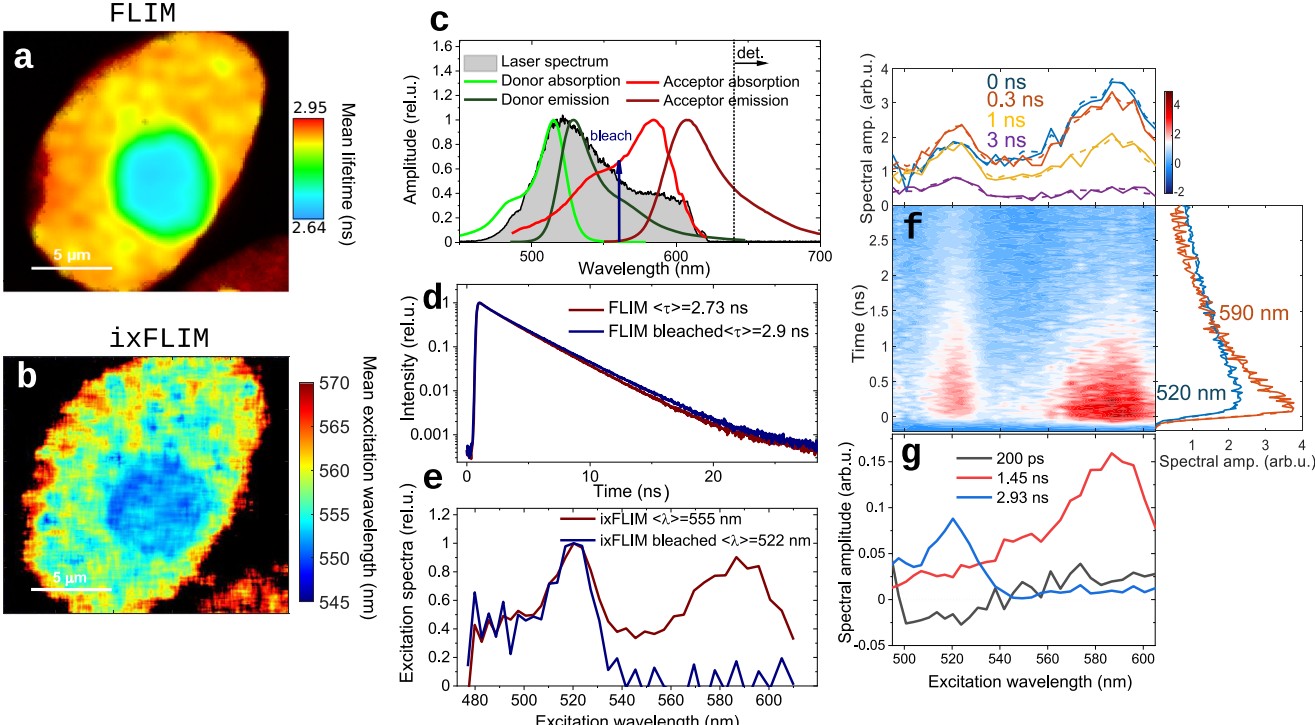

**Fig. 5 | Interaction of NPM in nuclei of HEK-293T cells by ixFLIM. a** FLIM image of a HEK-cell nucleus, excitation 488 nm, color codes mean emission lifetime. **b** time-integrated ixFLIM (ixFIM) image of the same nucleus as in (**a**), color codes mean excitation wavelength. **c** mVenus (green lines, shorter wavelength) and mRFP1 (red lines, longer wavelength) absorption (light shade) and emission (dark shade) spectra, overlaid by the broadband excitation pulse spectrum (black). The emission was detected through a 640 nm long-pass filter. The wavelength 560 nm used for acceptor bleaching is indicated as well (blue arrow). **d** FLIM decay averaged over the whole nucleus, before (dark red line) and after (dark blue line) acceptor bleaching. **e** ixFLIM spectra averaged from the whole nucleus and over the detection time, before (dark red) and after (dark blue) acceptor bleaching. The mean wavelengths were calculated from the spectral range 505 nm to 600 nm, as described in the Methods. **f** ixFLIM transient map of the whole nucleus, with horizontal (fixed time) and vertical (fixed wavelength) cuts at the given positions (solid lines), overlaid with the curves from global analysis (dashed lines). **g** Spectra associated to the three time components from the global analysis. The number of experimental repetitions is described in the Statistics and Reproducibility section in Methods.

red), slightly shorter in the nucleolus (Fig. 5b). The FLIM lifetime around 2.7 ns is shorter than that of the free donor (2.9 ns), and in ixFLIM the spectra of both the donor and acceptor are present. This is consistent with the donor-acceptor energy transfer taking place. There are, however, several possible other reasons for donor lifetime shortening in FLIM such as local cellular environment, and the donor spectrum in ixFLIM could originate from a large abundance of the free donor only. In FLIM, the connection to the acceptor is customarily shown by acceptor photo-bleaching. Doing so, we observe the expected recovery of the long donor lifetime (Fig. 5d, dark blue). In ixFLIM, the acceptor spectrum disappears (Fig. 5e, dark blue), confirming successful acceptor bleaching and excluding possible formation of an mRFP1 photo-conversion product[15]. In contrast to FLIM, in ixFLIM there is no need to bleach the sample, as the information about donor-acceptor interaction is present in the spectro-temporal correlation. In Fig. 5f the transient ixFLIM$(\lambda_t, t)$ map is shown, together with the spectro-temporal fit curves from global analysis. The minimum number of time components needed to describe the ixFLIM spectrum are three: 200 ps, 1.5 ns and 2.9 ns. The spectra of these components are depicted in Fig. 5g. In the 2.9 ns component we clearly recognize free mVenus absorbing around 520 nm, the 1.5 ns component corresponds to the mRFP1 acceptor absorbing around 585 nm. The fastest 200 ps component has a negative amplitude and the spectrum of the donor. As we have seen on the Cy3−Cy5 beads, this component thus corresponds to the FRET and proves the NPM interaction. In contrast to the Cy3−Cy5 constructs, not all donor-acceptor pairs are interacting in these cells, and there is a mix of donor-acceptor stoichiometries. As derived in the Methods section [Eq. (16)], the overall ixFLIM transient

spectrum can be described in this case as

$$ixFLIM(\lambda_\tau, t) = k_A \eta_A \phi_A \left\{ c_A \epsilon_A(\lambda_\tau) e^{-k_A t} + c_D^b \epsilon_D(\lambda_\tau) \frac{k_T}{k_T + k_D - k_A} \left( e^{-k_A t} - e^{-(k_T + k_D)t} \right) \right.$$
$$\left. + \frac{k_D \eta_D \phi_D}{k_A \eta_A \phi_A} \epsilon_D(\lambda_\tau) \left( c_D^f e^{-k_D t} + c_D^b e^{-(k_T + k_D)t} \right) \right\}.$$

$$(7)$$

Same as before, $k_A, k_D, k_T$ are the acceptor and donor decay rates and the transfer rate respectively, $c_A, c_D^b, c_D^f$ are the concentrations of acceptor, bound and free donor, $\phi_{A,D}$ are the respective quantum yields, $\eta_{A/D}$ the detection efficiencies of the acceptor/donor emission including filters, and $\epsilon_A(\lambda_\tau)$, $\epsilon_D(\lambda_\tau)$ are the donor and acceptor excitation spectra. We can therefore identify $k_D^{-1} = 2.9 \pm 0.1$ ns, $k_A^{-1} = 1.5 \pm 0.1$ ns and $(k_T + k_D)^{-1} = 0.18 \pm 0.05$ ns. Not only is the 0.2 ns component direct indication of FRET, but we have in the single measurement of the rather complex system all information needed to calculate the FRET efficiency $E = 1 - \frac{k_D}{k_D + k_T} = 0.93 \pm 0.03$ [Eq. (3) above]. Assuming, for lack of structure, random dye orientation and Förster radius of 55 Å[51], this implies distance of ∼35 Å between the donor and acceptor proteins. This is consistent with the oligomeric geometry, considering the distances between the NPM monomers in the pentamer ranging from 15 Å to 25 Å[54] and the mVenus/mRFP1 diameter of 25 Å[58].

As we have seen already from the ixFLIM and FLIM images (Fig. 5a, b), the lifetime is a bit shorter and spectrum bit more blue in the nucleolus compared to the surrounding nucleoplasm. We further analyze these regions separately as labelled in Fig. 6a. In Fig. 6b we contrast the FLIM decays and in Fig. 6c the excitation spectra of the

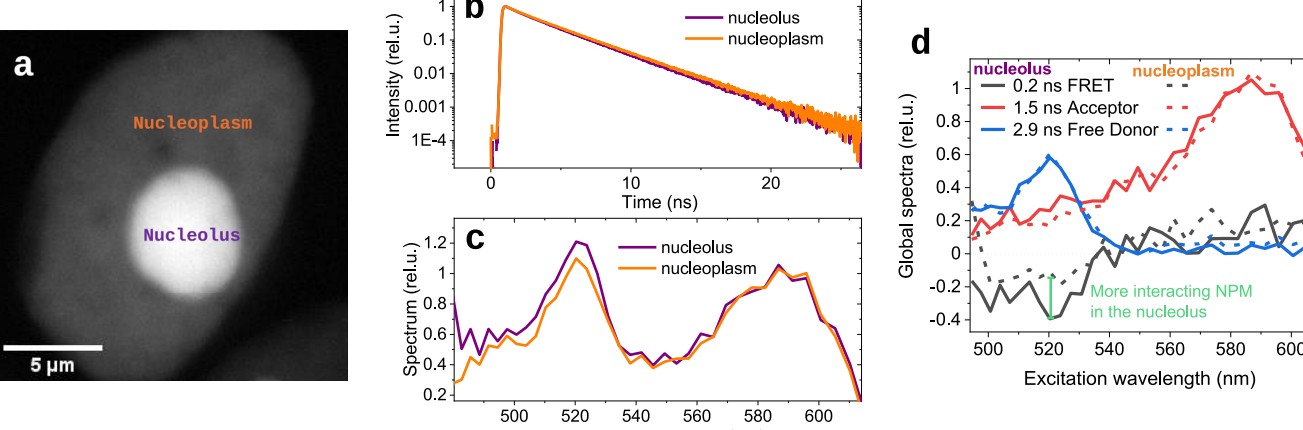

**Fig. 6 | Interaction of NPM in nucleoli and nucleoplasm. a** Fluorescence intensity image of the HEK-293T cell nucleus from Fig. 5, the nucleolus is much brighter than the surrounding nucleoplasm (image gamma value set to 0.5) due to the large concentration of NPM. **b** TCSPC decay curves from the donor FLIM for the nucleolus (purple) and nucleoplasm (orange) regions. **c** Time-integrated ixFLIM (ixFIM) excitation spectra for the nucleolus (purple) and nucleoplasm (orange) regions. The decay is a bit faster and spectrum has larger donor amplitude in the nucleolus due to a larger fraction of interacting NPM. **d** Global analysis of the nucleolus and nucleoplasm regions, spectra-associated to the three time components, normalized to the acceptor decay component. The only difference between the nucleolus (solid) and nucleoplasm (dashed) is the larger amplitude of the FRET-associated rise component (black), which has the same time constant of 0.2 ns.

nucleolus and the nucleoplasm, finding in the nucleolus a faster decay spectrum with higher donor spectrum amplitude. This difference could be caused either by more efficient FRET or by a larger fraction of interacting donor in the nucleolus. This is difficult to distinguish in the one-dimensional FLIM decay. In contrast, in ixFLIM the global analysis in Fig. 6d shows that the ratio of free donor to acceptor is the same in the nucleolus (solid line) and in the nucleoplasm (dashed line), while the FRET-associated component (black) has the same time constant (0.2 ns) but twofold amplitude in the nucleolus (green arrow). This means that there is the same type of oligomers of NPM in the nucleolus and in the nucleoplasm, but the fraction of interacting NPM is about two-times larger in the nucleolus. This agrees well with a shifted equilibrium due to a higher NPM concentration there, reflected by the larger fluorescence intensity as seen in Fig. 6a.

To summarize, FRET-ixFLIM allows direct, information-rich quantitative imaging of intracellular protein interactions within a single experiment, without need of tedious control bleaching-experiments which could even bias results e.g. by progressive sample evolution or degradation or by unwanted spectral photoconversion[2,15]. ixFLIM can also be used in combination with standard FLIM, to provide more constraints for the interpretation.

## Discussion

We have introduced a variant of FLIM with interferometric excitation, called ixFLIM. ixFLIM correlates the excitation spectrum and emission lifetime in each pixel (or voxel in 3D) of the fluorescence image. As we have shown, the additional dimension provides FLIM with the capability to assign and resolve various species in the fluorescence image. Established techniques such as global analysis can be used for this purpose. One of the most promising applications of ixFLIM is to FRET measurements. We have established a FRET-ixFLIM methodology to quantitatively measure distances and geometry of interacting molecular pairs, and to prove, isolate and image protein interactions within a single measurement. With ixFLIM, the FRET species can be reliably identified, with acceptor emission carrying information on both the donor and acceptor spectra even. This allows the use of FRET even for donors with small quantum yield or emission in unfavorable spectral regions. Crucially, donor-acceptor pairs with both fast and slow transfer can be imaged by ixFLIM within a single measurement. ixFLIM can be used even for the standard donor lifetime FLIM, shifting the excitation/detection region to the blue donor absorption/

emission, respectively. The spectral resolution can then be used to check for lack of sample integrity and lack of photo-converted species.

Related techniques exist in the literature. The closest one was recently applied to single-molecules[28]. Using the same interferometric two-pulse excitation, molecular excitation spectra were resolved and correlated with emission spectra and, alternatively, with the fluorescence decay. However, the decay was constant across the whole spectral range and no additional information beyond two ordinary one-dimensional datasets (excitation spectrum and fluorescence decay) was present. In the fluorescence spectroscopy, excitation spectra have been used to isolate species, see, chapter 16 in the book by Lakowicz and references therein[3]. However, to the best of our knowledge, the correlation between the excitation spectrum and the fluorescence lifetime has not been utilized, especially not in imaging. We note that already the time-integrated ixFLIM spectra (ixFIM) contain valuable information on the excitation, which we used in the spectral fits for the determination of FRET efficiency in the DNA samples. Established analysis of the excitation spectra such as spectral deconvolution[59] can thus be applied to ixFIM data well. Compared to sequential spectral scanning, the interferometric approach has advantages known from FTIR spectroscopy[60] and similar transform techniques such as speed, sensitivity and probing all wavelengths at the same time.

When extending a standard FLIM to ixFLIM, only excitation must be modified. ixFLIM can thus extend the capabilities of almost any FLIM microscope or imager (time or frequency domain, scanning or wide field). Regarding requirements for sample stability and acquisition time, ixFLIM is comparable to standard FLIM[5]. Since the data are dispersed along the additional spectral dimension, more photons have to be acquired. In practice, however, the excitation spectrum is modulated by stepping the delay $\tau$ between acquisition of single FLIM frames, which are in a standard FLIM often acquired sequentially for averaging. In this manuscript, typically 301 frames were acquired with scanning time 10−20 $\mu$s/pixel, i.e., about 0.6–1.3 s per 256 × 256 pixel image. This results in a measurement time of 3–7 min. The excitation light intensity was the same as for standard FLIM, the limitations being photon pile-up[61] and sample bleaching. The acquisition speed is determined by the required number of pixels, scanning speed, sample brightness and stability, laser repetition frequency, as well as by the required spectral and temporal range and resolution. Relaxing these, a

speedup by a factor of 10 can be easily achieved, so that an ixFLIM image can be acquired within tens of seconds. These measurement parameters are compatible with in vivo cell imaging.

There exist variants of FLIM with ordinary sequential scanning of excitation wavelength, or by using multiple pulse-interleaved lasers (PIE)[21–23]. The main difference between the interferometric scanning used in ixFLIM and sequential excitation scanning is that in the ixFLIM approach the excitation spectrum is dense and all excitation wavelengths interact with the sample simultaneously during the scan, while in sequential scanning only sparse wavelengths are sampled and the acquisition time increases proportionally to the number of excitation wavelengths. The Fourier-transform approach means that the spectral and temporal resolution are independent, and there is no experimental tradeoff between spectral resolution and excitation intensity, in contrast to spectral selection from a broadband source. We note here that the excitation intensity in the linear excitation regime is needed for ixFLIM. ixFLIM has a clear advantage in samples whose properties change during the measurement. This could be bleaching, sample movement (e.g., in live cells), or more complex processes such as the formation of photo-converted species. In the ordinary sequential scanning, one would obtain spectral distortions, since different wavelengths are scanned in different times. In ixFLIM, all points of the interferogram contribute to the spectra, so that the average spectrum will be observed. In the PIE approach the wavelengths are switched rapidly, but with more than a few lasers it quickly hits the pileup limit dictated by the required long time to amplitude converter time range and overall low effective repetition frequency, especially for longer emission lifetimes. On the other hand, in situations where the excitation spectra are known and FLIM at only two to three excitation wavelengths suffices, it might be more efficient to use standard FLIM with wavelength switching.

Considering suitable samples, ixFLIM requires comparable concentrations of the species to be resolved. This contrasts with FLIM, where an excess acceptor can be used to make the FRET-shortened donor decay more visible[62]. On the other hand, this shifts the interaction equilibria, which can be in some situations undesirable. In the FRET application, excess donor concentration works also for ixFLIM as long as the donor emission is sufficiently suppressed in the detection channel. For samples with complex composition (different species or with multiple fractions present), ixFLIM has a clear advantage due to the additional dimension, allowing to disentangle the otherwise one-dimensional FLIM decay according to the excitation spectrum.

ixFLIM measurement is more complex than standard FLIM, as it requires broadband pulsed light source and interferometric excitation scanning. ixFLIM has also larger photon budget due to the additional dimension to be resolved. We thus do not expect ixFLIM to replace FLIM, but rather envisage it to find use in applications where FLIM would be useful but faces problems. These are, e.g., FRET measurements with acceptors that photo-convert[15], with donors that feature multi-exponential decay kinetics[40], or donor/acceptor pairs whose lifetime or spectra depend on their microenvironment[9,10]. Next to the FRET studies, ixFLIM will find application where several species with similar lifetime and different spectra are present. A typical example is auto-fluorescence, typical for example in plant samples[63], another is application to high-throughput screening. ixFLIM can be also applied to sensing, where it extends the range of suitable probes to those whose spectrum shifts with the measured quantity. Because of the wealth of information from a single measurement, ixFLIM could be useful for precious or fragile samples. While the recent related single-molecule work by Thyrhaug et al.[28] suggests possible application to single molecules, due its relatively large photon budget we do not expect the ixFLIM to find application in single-molecule FRET studies[45]. While this manuscript showcases several biophysical applications of ixFLIM, other potential applications include e.g. optoelectronics and photovoltaics, forensics, or sensing applications. With both the advantages and limitations of ixFLIM in mind, we are convinced that ixFLIM will prove to be useful in a wide range of applications in life sciences and beyond.

## Methods

### FRET-ixFLIM description

In a typical FRET experiment, there is a pair of fluorescent molecules called donor and acceptor. After the excitation by the laser pulse, there will be a population of excited donors $P_D(0)$ and excited acceptors $P_A(0)$. If we assume rate kinetics, the equations for the following excited-state population dynamics are

$$\frac{dP_D(t)}{dt} = -k_D P_D(t) - k_T P_D(t)$$
$$\frac{dP_A(t)}{dt} = +k_T P_D(t) - k_A P_A(t). \tag{8}$$

Here, we have the rates $k_A, k_D$ for the decay of the isolated acceptor and donor, respectively, and the rate of the Förster transfer $k_T$. The solution is

$$P_D(t) = P_D(0)e^{-(k_D+k_T)t}$$
$$P_A(t) = P_A(0)e^{-k_A t} + P_D(0)\frac{k_T}{k_T+k_D-k_A}\left(e^{-k_A t} - e^{-(k_T+k_D)t}\right). \tag{9}$$

The initial populations are proportional to the excitation spectra of the acceptor and donor $\epsilon_A(\omega)$ and $\epsilon_D(\omega)$ and their concentrations in the sample $c_A$, $c_D$ (for brevity we drop the $\tau$ subscript by $\lambda_\tau$ in this section). First, we assume that a long-pass filter and sufficient spectral separation allow detection of the acceptor emission only, $FL(t) = k_A \phi_A \eta_A P_A(t)$, where $k_A$ is the rate of photon emission from the population $P_A$ and $\phi_A$ is the acceptor fluorescence quantum yield multiplied by the total detection efficiency. The transient excitation spectral map can then be expressed as

$$\text{ixFLIM}_A(\lambda,t) = k_A \eta_A \phi_A \left\{ c_A \epsilon_A(\lambda)e^{-k_A t} + c_D \epsilon_D(\lambda)\frac{k_T}{k_T+k_D-k_A}\left(e^{-k_A t} - e^{-(k_T+k_D)t}\right) \right\}. \tag{10}$$

This expression applies to our cyanine-dye-labelled beads, with $c_A = c_D$ by design (donor and acceptor on complementary strands of the double-stranded DNA). Clearly, in this case the mere presence of $\epsilon_D(\lambda)$ in the spectrum implies $k_T \neq 0$, i.e., directly reports on transfer. The acceptor spectrum has in ixFLIM always simple kinetics, decaying with the acceptor lifetime. The donor spectrum, however, exhibits the competition of donor-acceptor transfer and acceptor decay. For $k_T + k_D \gg k_A$ (often called normal kinetics), the signal first rises with the rate of $k_T + k_D$ and then decays with the lifetime of the acceptor, i.e., the rate $k_A$. For $k_T + k_D \ll k_A$ (often called inverted kinetics), the transferred excitation rapidly decays without building up acceptor population. In this case, the donor spectrum will at longer times decay with the shortened lifetime of the donor, i.e., rate $k_T + k_D$. Unlike in transient absorption, where the inverted kinetics is hard to observe due to the small buildup of $P_A$, both cases of transfer lead to a strong signal. The normal kinetics is exemplified by the cyanine-dye-labelled beads, where all signals clearly decay with the lifetime of the acceptor at longer times. Upon integration over the detection time, we obtain

$$\text{ixFLIM}_A(\lambda) = \int_0^\infty ix\text{FLIM}(\lambda,t)dt = \phi_A \eta_A \left\{ c_A \epsilon_A(\lambda) + c_D \epsilon_D(\lambda)\frac{1}{1+\frac{k_D}{k_T}} \right\}. \tag{11}$$

Here, we immediately recognize in the last fraction the transfer efficiency $E$ (Eq. (4) in the main text), getting

$$\text{ixFLIM}_A(\lambda) = \phi_A \eta_A \{ c_A \epsilon_A(\lambda) + c_D \epsilon_D(\lambda) E \}. \tag{12}$$

In case that appreciable ratio $\frac{\phi_D}{\phi_A}$ of the donor emission is detected as well, a term

$$\text{ixFLIM}_D(\lambda, t) = \phi_D \eta_D k_D c_D \epsilon_D(\lambda) e^{-(k_D + k_T)t} \tag{13}$$

has to be added to the expression Eq. (10) for ixFLIM$_A(\lambda,t)$ above, and the total signal is

$$\text{ixFLIM}(\lambda, t) = \text{ixFLIM}_A(\lambda, t) + \text{ixFLIM}_D(\lambda, t). \tag{14}$$

The implication is that the rise component in the donor spectrum is decreased by a factor of $\frac{\phi_D \eta_D k_D}{\phi_A \eta_A k_A}$. This is straightforward to understand since this fraction leaks through the filter and it is detected whether transferred to the acceptor or not. In the time-integrated signal (previously Eq. (12)), the donor leak translates to a correction to the extracted transfer efficiency $E$:

$$\text{ixFLIM}(\omega) = \phi_A \eta_A \left\{ c_A \epsilon_A(\lambda) + c_D \epsilon_D(\lambda) \left[ E + \frac{\phi_D}{\phi_A}(1 - E) \right] \right\}. \tag{15}$$

In more complicated samples, such as our HEK-293T cells with nucleophosmin labeled with mVenus and mRFP1, the situation can be more complex. The main complication is that there are multiple fractions of the emitting populations. In most cases, the population fractions will be independent of each other, i.e., each one will obey its dynamics. ixFLIM has the advantage of easily isolating other species $s$ than the desired donor-acceptor pair, since these will contribute as $k_s \phi_s \eta_s c_s \epsilon_s(\lambda) e^{-k_s t}$ and can be identified by their spectrum, their lifetime or both. The species $s$ can either be other molecules (such as autofluorescence of the cells), or a fraction of the donor/acceptor molecules with varying transfer efficiency and/or varying excited-state decay. For each pair of donor and acceptor, the dynamics in Eq. (9) apply, yielding two components to the overall dynamics. While this can complicate the analysis, the spectral separation is still helpful. In the acceptor spectral region, only the acceptor decay components are present. These can be fitted first and then fixed in the global fit. The spectra in the donor region then report on the interaction in case they exhibit rise (with the rate $k_T + k_D$) and/or decay with the lifetime of the acceptor obtained from the first fit. This is the case of our labelled HEK-293T cells, where the FRET presence indicates interaction and thus NPM oligomerization. The complete ixFLIM form for the labelled NPM with fractions of free donor and acceptor is

$$\begin{aligned} ixFLIM(\lambda, t) = k_A \eta_A \phi_A \Big\{ & c_A \epsilon_A(\lambda) e^{-k_A t} + c_D^b \epsilon_D(\lambda) \frac{k_T}{k_T + k_D - k_A} \left( e^{-k_A t} - e^{-(k_T + k_D)t} \right) \\ & + \frac{k_D \eta_D \phi_D}{k_A \eta_A \phi_A} \epsilon_D(\lambda) \left( c_D^f e^{-k_D t} + c_D^b e^{-(k_T + k_D)t} \right) \Big\}. \end{aligned} \tag{16}$$

Here, we separated the concentration of the donor to the bound $c_D^b$ and free $c_D^f$. For acceptor only the total concentration matters. Clearly, the amplitude of the rise component with the donor spectrum is decreased by the part of bound donor emission. On the other hand, the component with the acceptor decay time is insensitive to the emission from the donor.

## Optical setup

Our experimental setup is schematically shown and described in detail in Fig. S1 in the SI. Briefly, we use a custom-built ultra-broadband pulsed excitation coupled to a commercial microscope with FLIM. The pulses originate from a Ti:Sapphire oscillator (Chameleon Ultra II, Coherent), their repetition rate is decreased to 4 MHz or 8 MHz by a pulse picker (PulseSelect, APE) and the pulses are broadened to a range of about 400 nm to 1000 nm in a photonic crystal fiber (PCF). While our PCF is custom-made in collaboration with Photonics Bretagne, alternative fiber-based white light pulsed lasers with similar parameters are commercially available. The spectrum is restricted to the desired excitation window by a shortpass filter at the long-wavelength edge of the absorption (Thorlabs). The pulse pair for spectral modulation of the excitation is produced interferometrically, using a birefringent common-path interferometer called TWINS[64,65]. Our TWINS is custom-built using wedges from FOCTek mounted on a piezo stage (V-408, PI) that allows rapid and precise scanning of the delay time $\tau$, and features excellent both phase stability and beam pointing stability (see section S2 of the SI and Refs. 64,65). The pulse pairs are free-space coupled into the commercial scanning confocal fluorescence microscope (iX83 with FV1200 scanner, Olympus) where they are used to excite the sample with an apochromatic objective (Olympus UPlanSApo 10x, 0.4NA for the Oxo VI, Olympus UPlanSApo 40x, 0.95NA, for the Cy3–Cy5-labeled beads and water immersion Olympus UplanSApo 60xW 1.2NA for the HEK-293T cells). The fluorescence is separated by a long-pass filter (Thorlabs) and detected through a confocal pinhole in a de-scanned port by a cooled hybrid photomultiplier tube (PMA Hybrid 40, PicoQuant), counted by a TCSPC module (TimeHarp 260, PicoQuant). The intensity of excitation was attenuated such that the data collection rate was kept well below 5% of the pulse repetition rate to avoid pile-up in the photon counting. The acquisition was realized by SymphoTime software (PicoQuant), triggered by scanning software of the microscope (Olympus). The microscope scan is triggered by a custom-made program in Matlab (Mathworks), that coordinates the experiment and controls the other instruments as well. Reference excitation spectral profile at the entrance into the microscope is measured by a fiber spectrometer (SR-2, Ocean Insight).

## ixFLIM signal processing

The ixFLIM dataset ixFLIM$(x,y,t,\tau)$ is acquired in form of a stack of $N_\tau$ FLIM frames ($N_\tau$ is a number of $\tau$ steps, typically we use $N_\tau = 301$ and steps of 0.75 fs). The data 'hypercube' is exported from SymphoTime (PicoQuant) software and further processed by a custom-made Matlab (Mathworks) script. The data processing procedure is described in detail in section S2 of the SI. For each pulse delay $\tau$ (and thus FLIM frame), the modulated excitation laser spectrum Spec$(\lambda,\tau)$ is collected (see Fig. S1 for the setup). In the first step, the $\tau = 0$ point has to be found and the TWINS interferometer step size calibrated. This is done from the Fourier-transformed reference spectra sequence Spec$(\lambda, \tau) \rightarrow$ Spec$(\lambda, \lambda_\tau)$ in an automatized way. Once the $\tau = 0$ position and TWINS calibration is known, the ixFLIM stack can be Fourier-transformed into the frequency domain: ixFLIM$_\text{raw}(x, y, t, \omega_\tau) = $ Re $\int d\tau e^{i\omega_\tau \tau}$ixFLIM$_\text{raw}(x, y, t, \tau)$. In order to suppress the zero frequency, the mean of the dataset along $\tau$ is subtracted before taking the Fourier transform. The Fourier-transformed dataset is then converted from frequency to wavelength[66], including the Jacobian of the transform $d\omega = \frac{2\pi c}{\lambda^2} d\lambda$ that multiplies the converted spectrum by $\frac{1}{\lambda^2}$. Finally, the resulting full dataset ixFLIM$_\text{raw}(x, y, t, \lambda_\tau)$ is divided by the excitation laser spectrum Spec$(\lambda_\tau)$, obtaining the ixFLIM dataset ixFLIM$(x, y, t, \lambda_\tau) = \frac{\text{ixFLIM}_\text{raw}(x,y,t,\lambda_\tau)}{\text{Spec}(\lambda_\tau)}$. This is then used for imaging and further analysis, as described in the paper. Note, that the division by the laser spectrum can increase the noise, especially close to the edges of the laser spectrum. It is also possible to work with the non-normalized raw ixFLIM$_\text{raw}(x, y, t, \lambda_\tau)$ dataset.

The ixFLIM dataset ixFLIM$(x, y, t, \lambda_\tau)$ includes as its marginals standard FLIM: FLIM$(x,y,t) = \int d\lambda_\tau$ixFLIM$(x,y,t,\lambda_\tau)$, as well as excitation-spectrum imaging (interferometric excitation fluorescence imaging, ixFIM), ixFIM$(x, y, \lambda_\tau) = \int dt$ixFLIM$(x, y, t, \lambda_\tau)$.

Wherever mean lifetime is calculated, the first moment is meant:

$$\langle t \rangle = \frac{\int (t - t_0)\,\mathrm{FLIM}(t)\,\mathrm{d}t}{\int \mathrm{FLIM}(t)\,\mathrm{d}t},$$

where $t_0$ is the time when the excitation pulse arrives.

Analogously, the mean excitation wavelength is calculated as center of mass,

$$\langle \lambda_\tau \rangle = \frac{\int \lambda_\tau\,\mathrm{ixFIM}(\lambda_\tau)\,\mathrm{d}\lambda_\tau}{\int ixFIM(\lambda_\tau)\,\mathrm{d}\lambda_\tau},$$

where the integration is over the wavelength region covered by the excitation laser spectrum.

## Cy3−Cy5 DNA constructs

A set of five 27 nucleotide long DNA oligonucleotides was designed. The set consists of two complementary DNA strands, which can hybridize to form double-stranded DNA.

5´-GTGAGT[6]AAAGAGATACACATGGATTGAG-3´ oligonucleotide was biotinylated on its 5´-end and internally labeled with Cy3 fluorescent dye (fluorescence donor) at a position 6 (as marked by superscript number in the sequence above), this strand is referred to as D6. Next, four oligonucleotides with identical nucleotide sequence but different Cy5 (fluorescent acceptor) position were prepared: 5´-CT[2]CAATCCA[9]T[10]GTGTATCT[18]CTTACTCAC-3´ internally labeled with Cy5 at positions 2 (A2), 9 (A9), 10 (A10) and 18 (A18), respectively (as marked by superscript numbers in the sequence above). All oligonucleotides were synthesized and labeled by Integrated DNA Technologies, Inc. When hybridized together at 60 °C in GeneQ Thermal Cycler (Bioer Technology Co.) we obtained 4 dsDNA strands, which differed in fluorescent donor Cy3 and acceptor Cy5 distance: the distance was 4 bp for D6:A18, 12 bp for D6:A10, 13 bp for D6:A9 and 20 bp for D6:A2 dsDNA. Using biotinylated 5´-end of the D6 oligonucleotide, individual dsDNA strands were bound to Dynabeads® M-270 Streptavidin magnetic beads (Thermo Fisher Scientific) according to the standard protocol. After washing, labeled beads were resuspended in 50 mM Tris·HCl, pH 7.5, to the desired concentration. The beads were dropcasted on a microscope cover glass and covered by a layer of agarose to prevent Brownian motion.

## Cell cultivation, transfection and fixation

Adherent HEK-293T cells derived from human embryonic kidney cells were kindly provided by the Šárka Němečková laboratory, Institute of Hematology and Blood Transfusion, Prague, Czech Republic, which obtained them from J. Kleinschmidt, DKFZ, Heidelberg, Germany. The cells were cultured at 37 °C under standard cultivation conditions in RPMI growth medium (Sigma) supplemented with 10% FBS (Biochrom) under 5% $CO_2$ atmosphere. Plasmids for expression of NPMwt fused with mRFP1[62,67] and with mVenus[13] were amplified in E. coli and purified with PureYield Plasmid Miniprep System (Promega). The cells grown on a glass-bottom Petri dish were transfected using jetPrime transfection reagent (Polyplus Transfection) according to the manufacturer's protocol. The transfected cells were further grown for 24 h prior to cell fixation. The transfected cells were fixed with 4% paraformaldehyde and permeabilized by 0.5% Triton X-100 as described in ref. 2. After final wash, the cells were kept in the sterile PBS in 4 °C. Experiments with the fixed cells were done at room temperature.

## Statistics and reproducibility

All of the measurements were repeated several times, with highly comparable results. For each type of the samples, we prepared several batches, and on each batch we carried out multiple measurements. Specifically, For the Oxonol VI, we first measured the solutions with and without BSA, and then prepared two different micro-capillary samples. The sample shown here was measured six times, at different positions and focusing depths. For the DNA constructs, we prepared three batches of samples (donor/acceptor mixing, hybridization, attachment to beads), on the batch used in the paper we measured three different groups of beads. For the HEK-293T cells, two batches were grown, from the batch shown we measured 10 different cells. In all cases, the presented transient ixFLIM spectra were found to be highly reproducible across the repetitions, save for the absolute intensity given by the particular sample concentration. The presented results of the analysis are to be taken as representative ones, no statistics was performed.

## Reporting summary

Further information on research design is available in the Nature Portfolio Reporting Summary linked to this article.

## Data availability

The data used in this study are publicly available in the Zenodo repository, https://doi.org/10.5281/zenodo.13270499.

## Code availability

The code used to process the data and the code to calculate the Förster rate in the DNA samples are publicly available in the Zenodo repository, https://doi.org/10.5281/zenodo.13270499.

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

## Acknowledgements

We thank Adil Haboucha, Laurent Provino, Sébastien Claudot and David Méchin from Photonics Bretagne for providing the photonic crystal fiber for broadband white light generation. This project has received funding from the European Union's Horizon 2020 research and innovation programme under the Marie Skłodowska-Curie grant agreement No. 101030656. We also acknowledge the Czech Science Foundation grant No. 22-03875S.

## Author contributions

P.M. and P.H. conceived the project. P.M. designed and built the ixFLIM setup, performed the experiments, analyzed the data, and wrote the first version of the manuscript. D.S. designed and prepared the DNA oligonucleotide samples. A.H. prepared the HEK-293T cell samples. P.M. and P.H. acquired funding. All authors contributed to the final version of the manuscript, which they approved.

## Competing interests

The authors declare no competing interests.
