## [Peer Review File · Nature Communications]

Interferometric Excitation Fluorescence Lifetime Imaging MicroscopyREVIEWER COMMENTS

Reviewer #1 (Remarks to the Author):

In Maly et al, the authors introduce a new method, interferometric excitation FLIM (ixFLIM). The method is very interesting. However, some additional experiments are needed to verify the reliability of the method. In addition, some points need to be better clarified or discussed before the paper can be considered for publication.

Major points:

1) The method needs to be better described in the beginning of the paper (by Principle of ixFLIM). a) The authors should explain how they control the spectrum of the excitation pulse and how they remove the excitation laser from the detection path at the beginning. It is important to know that they are only looking at the red tail of the spectrum, much different than typical FLIM measurements with laser excitation. What the red-shifted detection window has for consequences on the detection efficiency also needs to be discussed. These points are not clearly explained until later in the text, if at all. It would also be interesting to show (or at least describe) what would happen when using a small band laser for excitation. b) The authors also need to explain in the principles section that the excitation spectrum is normalized to the laser spectrum (this information only comes at the end of the SI). c) The effect of directly exciting the acceptor molecules should also be discussed more in the text. For experiments with a single donor-acceptor pair, direct excitation of the acceptor molecules typically leads to no energy being able to be transferred from the donor. This would decrease the measured FRET efficiency. Do the authors observe such an effect? Do the authors observe excited state absorption? d) What happens if the authors change the spectrum of the excitation laser in an experiment (e.g. decrease the laser intensity in the spectral region of acceptor absorption)? How does this influence the results? Technically, with normalization, nothing should change. However, it will influence the amount of acceptor that is directly excited, impact the signal-to-noise and, if direct acceptor excitation is a problem for FRET, would also change the amount of FRET. Please discuss in detail.

As a side note, it would be very interesting if the authors would use emission filters to allow the detection of both donor and acceptor fluorescence in different detection channels.

Hence, the donor channel would yield the typical FLIM information with the donor lifetime and the acceptor channel the ixFLIM information. This would increase the robustness and sensitivity of the method. This is a suggestion for further development of the instrument and is far beyond the scope of the current manuscript.

2) Introduction: " For sensing and imaging applications, formation of aggregates can be recognized and eventually excluded,..." The formation of aggregates can be detected and removed with other methods as well. The method developed here is interesting and has some advantages. However, the authors should not oversell the technique. They should also discuss more of the drawbacks such as the slow data collection time ("7 minutes") which the authors try to sell as "fully compatible with in vivo cell imaging" and first mention in the discussion, and low sensitivity due to the limited fraction of the fluorescence emission spectrum detected by the setup.

3) DNA measurements: a) Donor only and acceptor only samples should be shown as a control. b) The authors should perform the DNA experiments with dyes separated far enough that FRET is not seen (as a control). This would be more interesting when the detection is sensitive to both dyes. d) The authors make assumptions about the flexibility of the fluorophores. The authors should measure the anisotropy to support their assumptions.

4) The analysis of the DNA data is problematic.

a) First of all, the FRET pair that is separated by only 4 bps is too close for normal FRET. Here, Dexter transfer can happen as well as quenching of the fluorophores. Hence, the adjustment of absorption spectra by " slightly adjusted multiplying by skewed Gaussians to fit the 4 base-pair" is not justified. b) The authors then attribute discrepancies in measured FRET efficiencies to the dipole geometry. This is too much speculation when using the measurement to demonstrate the reliability of the method. The authors need to repeat the DNA FRET experiments with a different dye pair. I would strongly recommend using the standard DNA pairs published in Hellenkamp et al (Nature Methods, 2018) where the expected FRET efficiencies are well documented.

5) For all determined rates, please include errors.

6) "This may be useful for studies of the local DNA geometry and flexibility." The geometry and flexibility of DNA has been well studied and documented by several groups using smFRET. (see e.g. Woźniak et al, PNAS 2008 105:18337).

7) Measurements of nucleophosmin (NPM) in HEK-293T cells. a) The situation measured here is much more complicated than discussed in the analysis. If NPM indeed forms pentamers or decamers, then several FRET efficiencies should be present. Hence, a single transfer rate is an oversimplification. From the data, it is clear that three components (with a single FRET rate) are sufficient for describing the data. However, the discussion should be adjusted to stressed more that this is an average rate and only an approximation. b) This brings up a second point. How sensitive is xiFLIM to a distribution of rates. In fact, the lifetime information is typically not subjected to averaging due to the short timescale of fluorescence. Hence, the lifetime decay is often used to detect a distribution of distances. c) As the authors are introducing a new method here, they should also determine the FRET efficiency on the same sample using traditional FLIM as a comparison. d) What I do not follow in the calculation of the bound and not-bound complexes is how direct excitation of the acceptor (at the wavelength region of acceptor absorption) is accounted for.

Minor points:

Principle of ixFLIM: "time resolution could be...". They authors should also explain here the timescale needed for imaging.

Figure 2: The panel order is very confusing. Please reformat the figure so that the panel order is easy to follow.

Equation 3: ϕ is used for quantum yield. It is confusing to use it here for the emission spectrum of the donor. They authors should also avoid using ϕ for the azimuthal angle in the SI. Using the same symbol for different things is confusing. With respect to nomenclature, R_0 is referred to as the Förster radius, not the "critical radius"

The idea of using the excitation spectrum to measure FLIM is not new. The authors should reference early work where this has been done (e.g. Eisinger 1969 Biochemistry 8:3902;

Eisenhawer et al 2001 Biochemistry 40:12321; Lakowicz Principles of Fluorescence, Third edition, Chapter 16.4.2, page 541ff).

"A standard FLIM image shows a uniform narrow distribution of lifetimes around 1.4 ns (inset in Fig. 3b), consistent with that of the Cy5 acceptor⁴¹, with small deviations easily explained by the known sensitivity of the fluorescence to the particular position along the DNA^{42,43}" The authors should confirm this independently with measurements on acceptor only samples.

Figure 3b. a) The ixFLIM images are dominated by the lifetime of the acceptor. It would be more useful for the FRET experiments to plot the lifetime of the rise on the donor signal in the acceptor channel. b) Do the magnetic beads introduce a fluorescence background? What do measurements with unlabeled magnetic beads look like?

"The presence of the donor Cy3 peak in the ixFLIM spectrum is in this case immediately a qualitative proof that the FRET takes place." This is true only if there is no spectral crosstalk of the donor into the acceptor channel. The authors should phrase this comment more precisely.

"...clearly show the delayed rise of the donor peak due to the FRET." Again, a bit more precision would be helpful. The standard FRET specialist will read Figure 3e as the donor and acceptor emission signals rather than the detected excitation signals. For ixFLIM experiments, one has to think differently and the authors should be more explicit to guide the readers.

Equation 6: This equation only holds if there is no detection of the signal from the donor and the fraction of excited acceptor molecules is low enough that the probability of excitation of both the donor and acceptor fluorophores on the DNA molecule is negligible. The authors should discuss this in more detail in the text.

Methods: FRET-ixFLIM description: a) The authors should describe what they mean by excitation spectra. Usually, epsilon is used for the absorption spectra. b) "In case that

appreciable ratio ϕ_D/ϕ_A of the donor emission is detected as well." Why does the ratio of quantum yields play a role here, or are the authors also incorporating the detection efficiency into the quantum yield? The quantum yield is a property of the fluorophore, not of the details of the setup. Please reword.

Methods: Optical Setup: a) The NA of the used objectives are missing. How sensitive is ixFLIM to the NA of the objective? b) "The intensity of excitation was attenuated such that the data collection rate was kept well below 5% of the 4 MHz repetition rate to avoid pile-up in the photon counting." 5% of 4 Mhz is well below 200 kHz. $200 \text{ kHz} * 20 \text{ us pixel time} \sim 4$ photons per pixel maximum. Perhaps the authors should elaborate on how they can determine the interference pattern with such low statistics per time bin.

SI: S3: spectral leak term not defined. Please define what you mean.

Reviewer #2 (Remarks to the Author):

This paper presents a variant of fluorescence lifetime imaging microscopy (FLIM) named ixFLIM that correlates the excitation wavelength with fluorescence decay. Rather than scanning the excitation wavelength, the authors utilize an interferometric technique. A broadband laser pulse is divided into two time-shifted, phase-stable replicas, thereby achieving spectral modulation. Data acquired as a function of the delay between pulses can be converted to a function of the wavelength via Fourier Transform. Consequently, each pixel contains the time-resolved fluorescence decay as a function of the excitation wavelength. The authors describe and analyze three experiments: Oxonol VI free and bound to BSA, FRET between Cy3 and Cy5 in DNA constructs, and HEK-293T co-transfected with nucleolar protein nucleophosmin bound to mVenus or mRFP1.

The presentation of the data contains several shortcomings. For example, residuals for the different fits are not displayed, making it impossible to ascertain the presence of systematic issues. Furthermore, uncertainties are absent from all fits throughout the text, preventing the reader from gauging the accuracy and precision of the method. Additionally, the organization of the panels within the figures is confusing, and many plots would benefit

from being presented in a stacked format. Examples of such figures include 2b,g and 3c,d,f,g; and 5d,e. This is not solely an aesthetic consideration, but rather a means of presenting the data that enables the reader to evaluate it more effectively.

The main claim of the authors is that the extra information available in this dataset enables resolving complex structures that are unsolvable with previously published methods. While I agree with the premise that the dataset produced by ixFLIM does contain valuable information, this paper does show its potential compared to other simpler, faster, commercially out of the box, and well established FLIM variants. For example: how does it compare with measurements performed using two or three lasers using pulsed interleaved excitation? While it contains much less information than ixFLIM, it might be enough to resolve the experimental examples described here. If this is true, what is the precision and accuracy of each one? When is it worth it to use ixFLIM? These questions must be addressed quantitatively; which could be done, for example, by slicing the ixFLIM dataset. Additionally, a spectroscopically more complex system which can only be resolved by ixFLIM must be shown.

Also, how does it compare with FLIM using sequential scanning of the excitation wavelength, which is available in some microscopes equipped with a white light pulsed laser and an acousto-optical beam splitter?

As a final, minor note, the statement "The measurements parameters are fully compatible with in vivo cell imaging" is way too broad. The authors assert that it takes 7 minutes to capture a dataset. A prevalent application of FLIM in cellular imaging is for monitoring the state of biosensors, molecular interactions, and post translational modifications within signaling. In many of these instances, 7 minutes encapsulates the entire process duration, and therefore this technique in its current form is too slow.

Reviewer #3 (Remarks to the Author):

The manuscript titled "ixFLIM: Inetreferometric Excitation", by X et al. describes a technique for fluorescence lifetime imaging that the authors call ixFLIM. The technique

allows the excitation of fluorophores at a continuous range of wavelengths, instead of using a few discrete ones which have been the best practice so far. It has the potential to separate more fluorophore species in the sample than what is possible with discrete excitation. In practice, the authors use a broadband coherent light source generated from a femtosecond Ti:sapphire oscillator, and interfere them at the sample with a specific time delay generated using an interferometric device. The variation of the delay generates different excitation profiles, and this can be transformed into 2D data which encoded both the excitation and the emission spectra of all the fluorophores present in the sample.

The manuscript is well written and describes several example data which makes the use of ixFLIM clear. However, before I can recommend publishing the manuscript in Nature Comm, several points need to be addressed. These are:

- 1) The major concern I have is about the practicality of the technique. A multiwavelength excitation should be able to do, in practice, faster and more reliable chemical diagnostics. However, the increased instrumental complication and the time taken to record the data are apparent negative points of this method. The manuscript will benefit from a direct comparison of all the aspects between conventional FLIM and ixFLIM.
- 2) Similarly, a comparison between the established techniques and this new one should be performed on some real or standard samples.
- 3) It is not clear how the background from the excitation which leaks into the emission channel and how it can be eliminated.
- 4) Line 239: How would this technique compare with the standard spectral deconvolution technique?
- 5) Line 355: 7 min seems to be a long observation time

Minor points:

- 1) Page 16: "critical" is not typically used
- 2) Fig. 3 caption: "weighed sum" should become "weighted sum"

Reply to Reviewer Comments on Manuscript NCOMMS-23-56256A

Title: "ixFLIM: Interferometric Excitation Fluorescence Lifetime Imaging Microscopy"

Authors: Pavel Malý, Dita Strachotová, Aleš Holoubek and Petr Heřman

We are content that all three reviewers find the ixFLIM method interesting and recommend publication after appropriate revisions. We hereby wish to thank them for their time taken for the detailed evaluation of our manuscript.

Based on the reviewer and editorial comments, we have thoroughly revised the manuscript. We left out some specific details and shifted the emphasis to the comparison with FLIM. We carried out a number of additional ixFLIM and standard FLIM measurements:

Cy3-Cy5 DNA FRET samples:

- conventional donor lifetime based FLIM of all samples
- standard FLIM and ixFLIM measurements of donor-only and acceptor-only beads, separate and adjacent
- standard FLIM and ixFLIM measurement of unlabeled beads

NPM in HEK cells:

- conventional donor-lifetime-based FLIM and ixFLIM measurements on the same cells

These measurements serve as a thorough characterization of the DNA FRET constructs, validate the ixFLIM technique, and allow us to directly compare the ixFLIM and FLIM methods on the same samples. We have significantly rewritten parts of the manuscript to elaborate on the relation of ixFLIM to conventional FLIM and other methods. In addition, we have added a discussion section about the envisaged use of ixFLIM.

Below, we describe our response to the reviewer comments point by point. The original comments are in *italics*, our response in upright font, and the particular actions taken in the manuscript in **boldface**.

REVIEWER #1 (REMARKS TO THE AUTHOR):

In Maly et al, the authors introduce a new method, interferometric excitation FLIM (ixFLIM). The method is very interesting. However, some additional experiments are need to verify the reliability of the metho. In addition, some points need to be better clarified or discussed before the paper can be considered for publication.

We are content that the reviewer finds our method interesting. We have carried out the additional experiments and rewrote parts of the manuscript text, as described below.

Major points:

1) The method needs to be better described in the beginning of the paper (by Principle of ixFLIM). a) The authors should explain how they control the spectrum of the excitation pulse and how they remove the excitation laser from the detection path at the beginning. It is important to know that they are only looking at red tail of the spectrum, much different than typical FLIM measurements with laser excitation. What the red-shifted detection window has for consequences on the detection efficiency also needs to be discussed. These points are not clearly explained until later in the text, if at all.

a) We have extended the diagram in Fig. 1, showing the excitation and detection spectral windows, together with the excitation and emission spectra of the exemplary donor/acceptor fluorescent proteins. We have also included in the introduction a description of how the detection is separated from the excitation spectrally using spectral filters.

In addition, we have added in the SI a figure illustrating the interferometric control of the excitation, the resulting interferograms and the ixFLIM spectra. We copy the figure here for convenience.

Figure S3 Demonstration of the ixFLIM signal processing. Left: image of the bp20 beads binned down to 64x64 pixels. Top: TCSPC time-integrated interferogram of a single ball (4x4 pixels), the Fourier transform of which produces the raw spectrum (blue), division of that by the laser spectrum (black) leads to the ixFLIM excitation spectrum (green). Bottom: spatially integrated image produces the time-time correlation map, on x axis is the interferometric time τ , on the vertical axis the TCSPC time delay. Fourier transformation along τ produces the ixFLIM transient map.

It would also be interesting to show (or at least describe) what would happen when using a small band laser for excitation.

To obtain the spectral information, broadband excitation covering the absorption peaks is needed in ixFLIM. ixFLIM doesn't work with narrow band lasers.

We included in the revised manuscript a sentence **“the pulses must be broadband to spectrally cover the absorption of all fluorophores of interest”**

b) The authors also need to explain in the principles section that the excitation spectrum is normalized to the laser spectrum (this information only comes at the end of the SI).

We explicitly mention the normalization to the reference excitation spectrum in the ixFLIM principle section on page 2 of the revised manuscript: **“After conversion from frequency ω_τ to wavelength λ_τ , the data are divided by the known imprinted laser excitation spectrum along λ_τ ,”**. No other normalization was done.

c) The effect of directly exciting the acceptor molecules should also be discussed more in the text. For experiments with a single donor-acceptor pair, direct excitation of the acceptor molecules typically leads to no energy being able to be transfer from the donor. This would decreased the measured FRET efficiency. Do the authors observe such an effect? Do the authors observe excited state absorption?

We indeed do excite the acceptor directly. However, ixFLIM also resolves the excitation spectrum, in which the donor and acceptor are spectrally separated. It is actually the ratio between the donor and acceptor excitation spectra that presents one way to determine the transfer efficiency from the ixFLIM data, according to Eq. 6. Unlike in spectrally resolved emission, in ixFLIM the energy transfer is visible as a rise of the spectrum of the donor.

We specify that the acceptor detection contrasts with standard FLIM on page 6: “in contrast to standard donor-lifetime FLIM, we detect emission from the acceptor ... only”.

The experiment is done on a large ensemble of fluorophores. It is linear in the excitation intensity, far from saturation, so we cannot observe any nonlinear effects such as excited state absorption. In this regime, the probability of having both donor and acceptor of the same FRET pair excited is negligible.

d) What happens if the authors change the spectrum of the excitation laser in an experiment (e.g. decrease the laser intensity in the spectral region of acceptor absorption)? How does this influence the results? Technically, with normalization, nothing should change. However, it will influence amount of acceptor that is directly excited, impact the signal-to-noise and, if direction acceptor excitation is a problem for FRET, would also change the amount of FRET. Please discuss in detail.

As the reviewer points out, the spectral shape dependence is removed by division of the measured dataset by the laser spectrum. For this, it is important to measure the spectrum correctly, we do this for each frame by taking the reflection of the beam entering the microscope and focusing it via a 10x objective into a multimode fiber, as we describe in the Methods section.

The spectral division can be a source of noise if the spectral amplitude is very weak. This happens inevitably at the edges of the laser spectrum, which thus have to be broad enough to cover the relevant spectral region. As long as the spectral shape is stable and smooth without significant structure and deep modulations, the ixFLIM results are independent of the excitation spectrum and so are the FRET results.

In the Methods section, we inserted the following: “Note, that the division by the laser spectrum can increase the noise, especially close to the edges of the laser spectrum. It is also possible to work with the non-normalized raw ixFLIM_{raw}(x, y, t, λ_τ) dataset.”

As we explain in the answer to the previous point, the direct acceptor excitation is not a problem and is accounted for in the analysis of the ixFLIM data.

As a side note, it would be very interesting if the authors would use emission filters to allow the detection of both donor and acceptor fluorescence in different detection channels. Hence, the donor channel would yield the typical FLIM information with the donor lifetime and the acceptor channel the ixFLIM information. This would increase the robustness and sensitivity of the method. This is a suggestion for further development of the instrument and is far beyond the scope of the current manuscript.

We thank the reviewer for this suggestion. While this is in principle possible, it would require structuring the excitation spectrum such that there is a detection ‘hole’ for the donor emission. With a special set of spectral filters, this should be possible. However, since this is dye-specific, we think a better strategy to detect the donor would be to choose donor-acceptor pairs with smaller spectral separation and thus overlapping partially emission. Yet another option is to restrict the spectral range to the donor absorption and emission and use ixFLIM to monitor the donor spectrum. **We mention this possibility in the Discussion on page 12 in the revised manuscript.**

2) Introduction: " For sensing and imaging applications, formation of aggregates can be recognized and eventually excluded,..." The formation of aggregates can be detected and removed with other methods as well. The method developed here is interesting and has some advantages. However, the authors should not oversell the technique. They should also discuss more of the drawbacks such as the slow data collection time ("7 minutes") which the authors try to sell as "fully compatible with in vivo cell imaging" and first mention in the discussion, and low sensitivity due to the limited fraction of the fluorescence emission spectrum detected by the setup.

We do not claim that ixFLIM is always better than other methods such as conventional FLIM, and the detection of an aggregate formation was just an example of use. It is true that the acquisition of a high-resolution ixFLIM image typically takes couple of minutes, in order to provide reasonable SNR in enough image pixels. However, this is comparable with FLIM. The 7-min acquisition is an example from the NPM measurement with 256x256 pixels and detailed interferogram sampling. The ixFLIM measurement time depends on factors such as sample brightness, scanning speed, number of pixels, spectral range, laser repetition rate, and desired spectral resolution and time resolution. On bright samples with reduced number of spatial pixels, one can easily measure ixFLIM in tens of seconds. This is similar to the standard FLIM, which needs longer accumulation time to well characterize dim samples. **In the Discussion section, we have included a paragraph discussing the advantages and disadvantages of ixFLIM compared to conventional FLIM and other methods, listing examples in which using ixFLIM is of merit.**

3) DNA measurements: a) Donor only and acceptor only samples should be shown as a control. b) The authors should perform the DNA experiments with dyes separated far enough that FRET is not seen (as a control). This would be more interesting when the detection is sensitive to both dyes. d) The authors make assumptions about the flexibility of the fluorophores. The authors should measure the anisotropy to support their assumptions.

We have carried out a number of additional measurements:

a) We have obtained the biotin-tagged oligonucleotides with acceptor only, and measured ixFLIM spectra of donor only and acceptor only beads. We show Fig. S7 (excitation spectra) and Fig. S12 in the SI and copy them here for convenience

Figure S7 Decomposition of the excitation spectra into those of the donor and acceptor. Left: separately measured excitation spectra of the Cy3 donor (blue) and Cy5 acceptor (orange). Right: the fits of the raw (not divided by the laser spectrum) ixFLIM (ω_τ) spectra (blue) by the (laser-spectrum multiplied) weighted sum of the donor and acceptor excitation spectra (orange) according to Eq. 6 in the main text. Fit residues are shown in black dotted lines.

Figure S12 Beads colored by Cy3 only. From left: the fluorescence microscope image, ixFLIM transient map (detection 600 nm long-pass filter), two components obtained from the global analysis fit, fit residue.

b) Furthermore, we have carried out a measurement in which the donor and acceptor are so far separated that no FRET can take place. Due to a larger donor-labelled DNA concentration, we do see in ixFLIM a very faint donor emission in the data. The results are discussed in the main text, are shown in Fig. S16 in the SI and copied here for convenience

We thank the reviewer for these useful suggestions.

Figure S16 Measurement with two beads 0.5 micrometer apart, one labelled by Cy3 donor and one by Cy5 acceptor. Top left: FLIM image shows the donor bead only. Top middle: ixFLIM measurement, intensity image showing strong acceptor bead and weak donor bead. Top right: FIM image clearly distinguishes the donor and acceptor based on their excitation spectrum. Bottom left: FLIM decay curve which agrees with the donor lifetime from global analysis in Fig. S11. Bottom right: ixFLIM transient map, two time components and fit residue. The ixFLIM shows almost exclusively the acceptor, with very weak donor spectrum present, but with no negative (i.e., rising) time component. There is thus no FRET taking place, as expected.

d) While we agree that measuring anisotropy could be very interesting from the point of possible the Cy3 donor photo-isomerisation dynamics, we conclude that this would complicate the analysis that already contains many datasets even further, without being directly related to the FRET efficiency. Instead, **we have included a reference to the work by Sanborn et al., Ref. 40** that discusses the sensitivity of Cy3 geometry to local environment, and we also included a **comparison with the work by Cunningham et al., Ref. 38**, where the structural disorder is discussed in detail. Concerning the DNA fragments, no prior explicit assumptions were made. We assumed and evaluated mean structural DNA disorder without any kinetic aspects or references to conformational distributions.

4) *The analysis of the DNA data is problematic.*

We have significantly extended the analysis of the Cy3-Cy5 DNA data, adding additional measurements and comparison to literature, especially to the work by Cunningham et al., Ref. 38.

a) *First of all, the FRET pair that is separated by only 4 bps is too close for normal FRET. Here, Dexter transfer can happen as well as quenching of the fluorophores. Hence, the adjustment of absorption spectra by "slightly adjusted multiplying by skewed Gaussians to fit the 4 base-pair" is not justified.*

4 base pair distance amounts to 1.36 nm. We agree that this could be too close for pure Förster transfer, especially when considered without generalization beyond the dipole-dipole coupling term. This has been also concluded by Cunningham et al, Ref. 38. However, the same authors and other studies have measured near- 100% FRET efficiency at a distance of 4 bp and even closer. From the point of our work, we determine the transfer efficiency and do not aim to distinguish the particular transfer mechanism at the short distance. **We added explicit discussion of the limitation of Förster theory in the DNA FRET section of the manuscript on page: "For the 4 bp dye separation the Förster theory with dipole-dipole coupling might not hold anymore. There is, however, no acceptor**

fluorescence quenching at this distance and near-100% efficient transport, same as expected from Förster theory³⁸”.

Regarding the adjustment of the absorption spectra, **we have replaced these adjusted spectra from literature by directly measured Cy3 donor and Cy5 acceptor excitation spectra** shown in Fig. S7 and in the answer to the point 3b) above. We used these unmodified spectra for the spectral fits of the ixFLIM data in Fig. 3g, Fig. 4b and Fig. S7.

b) The authors then attribute discrepancies in measured FRET efficiencies to the dipole geometry. This is too much speculation when using the measurement to demonstrate the reliability of the method. The authors need to repeat the DNA FRET experiments with a different dye pair. I would strongly recommend using the standard DNA pairs published in Hellenkamp et al (Nature Methods, 2018) where the expected FRET efficiencies are well documented.

While we maintain that there can be systematic differences based on the method of acquisition, we agree that, within the experimental error, these are too small to be determined reliably. We thus decided to discuss such details out of the main text. Instead, in the revised version we focus on the agreement of the measured efficiencies with each other, with standard FLIM, with theory, and with existing literature.

We have carried out additional standard donor-FLIM measurements of the Cy3-Cy5 DNA samples. The obtained efficiencies agree well with those extracted from ixFLIM, both by spectral fit and by the global analysis. Furthermore, **we quantitatively compare our results to those from Cunningham et al., Ref. 38,** showing a good agreement. Taking the Förster radius determined by Cunningham et al., we re-calculate our theoretical model, the results of which agree with the experiment as well. **We summarize and compare the efficiencies in the revised Figure 4 and Table 1 of the main text, which we copy here for convenience.**

Figure 4 | ixFLIM-based FRET imaging. a) FRET efficiency dependence on the donor-acceptor distance along the DNA. Values obtained by ixFLIM global analysis (cyan), ixFLIM spectral fit (red), donor lifetime FLIM (black) and by calculation (green dashed line) are compared between themselves and with values from literature, Ref. 38 (purple). To illustrate the sensitivity to geometry, the dependence for a hypothetical isotropic molecular orientation is shown as well. The error bars represent standard deviation, for the FLIM data the error bars reach 25% of the values (see Table 1) and were omitted for clarity. b) ixFLIM image of a mixture of beads with Cy3-Cy5 donor-acceptor pairs separated by 4 bp and 20 bp. The image is color-coded by FRET efficiency extracted in each pixel from the ratio of the donor and acceptor excitation spectra. c) The same sample measured by FLIM, determination of the bp04 lifetime is unreliable due to very low intensity and short lifetime caused by the FRET efficiency close to 1. Two pairs of beads are highlighted in both images with FRET efficiency of $E=0.7$ (green, 20 bp) and $E=1$ (red, 4 bp).

Table 1 FRET efficiency as a function of Cy3-Cy5 position along the DNA: summary of results.

Cy3-Cy5 distance	Signal rise fit	Spectral fit	FLIM	Theory:
4 bp (1.36 nm)	0.91±0.07	0.99±0.11	0.92±0.27	1.0
12 bp (4.08 nm)	0.81±0.03	0.81±0.09	0.79±0.41	0.78
13 bp (4.42 nm)	0.73±0.05	0.68±0.09	0.69±0.26	0.69
20 bp (6.8 nm)	0.68±0.06	0.72±0.09	0.66±0.19	0.71

We moved the discussion of the slight differences in the theoretical efficiency extracted from spectra and kinetics into the revised SI, section S3.

Regarding the possibility of using another dye pair, we contacted the IBA GmbH company that produced the nucleotide samples for the suggested study of Hellenkamp et al.. We were told that they do not make these anymore. We are content that the idea of using DNA-bound dyes for known distances has been used as a benchmark, i.e., on the same application as in our work. **We have included the reference to Hellenkamp et al in the manuscript, Ref. 45.** Since the Cy3 and Cy5 dyes are very commonly used in DNA labelling applications and their interaction has been studied in detail, **we have decided to stay with these dyes** and carry out the additional measurements, analyzes and comparison to literature to verify the ixFLIM method.

To verify the applicability of ixFLIM to FRET on these samples, we measured standard FLIM as well, the obtained FRET efficiencies agree between the techniques. **We show the standard FLIM fits in Fig. 3h and in Fig. S15 in the SI** and copy them here for convenience

Figure S15 FLIM measurements on the beads with increasing base-pair distance. Shown is always FLIM decay curve (red), its fit (pink) and fit residue (red dotted), and FLIM decay with bleached acceptor (black), its fit (grey) and the fit residue (black dots). The fits were done globally using three exponentials with fixed relative amplitudes, re-convoluted with the IRF decomposed into two Gaussians.

5) For all determined rates, please include errors.

We have calculated the uncertainties for all the calculated rates and efficiencies. **We present them with all reported values in the revised manuscript.**

6) *"This may be useful for studies of the local DNA geometry and flexibility." The geometry and flexibility of DNA has been well studied and documented by several groups using smFRET. (see e.g. Woźniak et al, PNAS 2008 105:18337).*

To provide the context and highlight the importance of dye labelling for DNA structure determination, **we have expanded the reference list by Refs 42,43,53, including the suggested smFRET reference.**

7) *Measurements of nucleophosmin (NPM) in HEK-293T cells. a) The situation measured here is much more complicated than discussed in the analysis. If NPM indeed forms pentamers or decamers, then several FRET efficiencies should be present. Hence, a single transfer rate is an oversimplification. From the data, it is clear that three components (with a single FRET rate) are sufficient for describing the data. However, the discussion should be adjusted to stressed more that this is an average rate and only an approximation. b) This brings up a second point. How sensitive is ixFLIM to a distribution of rates. In fact, the lifetime information is typically not subjected to averaging due to the short timescale of fluorescence. Hence, the lifetime decay is often used to detect a distribution of distances. c) As the authors are introducing a new method here, they should also determine the FRET efficiency on the same sample using traditional FLIM as a comparison. d) What I do not follow in the calculation of the bound and not-bound complexes is how direct excitation of the acceptor (at the wavelength region of acceptor absorption) is accounted for.*

a) We completely agree that the situation is complex in the oligomerizing NPM samples. **We have therefore reworked the section**, focusing on proving the NPM interaction by FLIM and ixFLIM. In the text, **we discuss the actual presence of several transfer rates, and introduce our description with one average transfer rate as an effective approximation.**

b) ixFLIM relies on the same process of time-resolving the fluorescence emission as FLIM, extending it by spectrally resolving the excitation. As such, the sensitivity of ixFLIM to a distribution of rates is the same or better as FLIM because it can resolve fast transfer as well. Attempting to distinguish the time components in the NPM measurement is, however, difficult due to the large fraction of free donor, and beyond the scope of this work, in which the measurement is used as a demonstration.

c) We have carried out additional standard FLIM measurement on the NPM-HEK sample. In the new Fig. 5 in the main text, shown here for convenience, **we directly compare FLIM and ixFLIM on a single nucleus of a HEK cell**, showing a good agreement and a degree of complementarity in proving the FRET and demonstrating larger fraction of interacting NPM in the nucleolus compared to the nucleoplasm.

Figure 5 | Interaction of NPM in nuclei of HEK-293T cells by ixFLIM. a) FLIM image of a HEK-cell nucleus, excitation 488 nm, color codes mean emission lifetime. b) time-integrated ixFLIM (ixFLIM) image of the same nucleus as in a), color codes mean excitation wavelength. c) mVenus (green lines) and mRFP1 (red lines) absorption (light shade) and emission (dark shade) spectra, overlaid by the broadband excitation pulse spectrum (black). The emission was detected through a 640 nm long-pass filter. The wavelength 560 nm used for acceptor bleaching is indicated as well (blue arrow). d) FLIM decay averaged over the whole nucleus, before (dark red line) and after (dark blue line) acceptor bleaching. e) ixFLIM spectra averaged from the whole nucleus and over the detection time, before (dark red) and after (dark blue) acceptor bleaching. f) ixFLIM transient map of the whole nucleus, with horizontal (fixed time) and vertical (fixed wavelength) cuts at the given positions (solid lines), overlaid with the curves from global analysis (dashed lines). g) Spectra associated to the three time components from the global analysis.

d) The calculation of the bound/unbound fractions was too detailed for this proof-of-principle manuscript. Moreover, it did not reflect the presence of multiple rates due to the pentameric NPM oligomerization. **We have removed the calculation of bound/unbound donor fractions** in the revised manuscript. Instead, we focus on the comparison with FLIM and we determine the difference in the relative fraction of interacting NPM in the nucleolus and nucleoplasm. **We added a new Fig. 6 illustrating the advantage of ixFLIM in imaging this difference**, shown here for convenience.

Figure 6 | Interaction of NPM in nucleoli and nucleoplasm. a) Fluorescence intensity image of the HEK-293T cell nucleus from Fig. 5, the nucleolus is much brighter than the surrounding nucleoplasm (image gamma value set to 0.5) due to the large concentration of NPM. b) TCSPC decay curves from the donor FLIM for the nucleolus (purple) and nucleoplasm (orange) regions. c) ixFLIM excitation spectra for the nucleolus (purple) and nucleoplasm (orange) regions. The decay is a bit faster and spectrum has larger donor amplitude in the nucleolus due to larger fraction of interacting NPM. d) Global analysis of the nucleolus and nucleoplasm regions, spectra-associated to the three time components, normalized to the acceptor decay component. The only difference between the nucleolus (solid) and nucleoplasm (dashed) is the larger amplitude of the FRET-associated rise component (black), which has the same time constant of 0.2 ns.

Minor points:

Principle of ixFLIM: "time resolution could be...". They authors should also explain here the timescale needed for imaging.

Here, we discuss temporal resolution of the fluorescence decay. **We have added thorough discussion of the timescale needed for the imaging in the Discussion section on page 13 of the manuscript.**

Figure 2: The panel order is very confusing. Please reformat the figure so that the panel order is easy to follow.

We have re-designed Fig. 2 to follow the order of discussion in the text.

Equation 3: phi is used for quantum yield. It is confusing to use it here for the emission spectrum of the donor. They authors should also avoid using phi for the azimuthal angle in the SI. Using the same symbol for different things is confusing. With respect to nomenclature, R0 is referred to as the Förster radius, not the "critical radius"

In the revised version of the manuscript, we **keep the ϕ for quantum yield and use α for the angle. We rephrased the R0 as Förster radius, and revised the relevant formulae accordingly.**

The idea of using the excitation spectrum to measure FLIM is not new. The authors should reference early work where this has been done (e.g. Eisinger 1969 Biochemistry 8:3902; Eisenhawer et al 2001 Biochemistry 40:12321; Lakowicz Principles of Fluorescence, Third edition, Chapter 16.4.2, page 541ff).

We agree that the idea to use the excitation spectrum to disentangle species exists in literature. However, we maintain that, to the best of our knowledge, it has not been applied in combination with lifetime measurement and imaging, i.e. FLIM. The suggested references discuss the application to FRET measurement in volume samples, without any temporal or spatial resolution. **We have included a discussion of the relation of ixFLIM to other techniques in the Discussion section on pages 11-12 of the main text, where we cite the Lakowicz book (Ref. 3) with the relevant references therein.**

"A standard FLIM image shows a uniform narrow distribution of lifetimes around 1.4 ns (inset in Fig. 3b), consistent with that of the Cy5 acceptor⁴¹, with small deviations easily explained by the known sensitivity of the fluorescence to the particular position along the DNA^{42,43}" The authors should confirm this independently with measurements on acceptor only samples.

We have confirmed this by analysing the ixFLIM data in the acceptor excitation region only, which can be fitted with a single-component decay with 1.35 ns time constant. We have added the fit into the SI (Fig. S11) and copy it here for convenience.

Figure S11 ixFLIM in the acceptor region only (from the b13 data). Since the uphill FRET is insignificant due to the large energy gap, this is equivalent to the free acceptor. Transient map, single time component used for the global fit, residue.

Figure 3b. a) The ixFLIM images are dominated by the lifetime of the acceptor. It would be more useful for the FRET experiments to plot the lifetime of the rise on the donor signal in the acceptor channel. b) Do the magnetic beads introduce a fluorescence background? What do measurements with unlabeled magnetic beads look like?

a) The purpose of this picture is to show how the ixFLIM dataset looks as detected in the acceptor channel, before any analysis. **We have thus left the figure panel** as it is, only moving the lifetime distribution into the SI since it is not important. The equivalent of the rise time is the FRET efficiency $E = k_T / (k_D + k_T)$, **which we plotted in Fig. 4b** in the revised version, **directly compared to donor FLIM in Fig. 4c**. The figure can be found by the answer to point 4b above.

b) We have also measured the unlabelled beads. They show very weak emission in the green (donor emission region) and basically no emission in the red (acceptor emission region). **We have included these in Fig. S14 in the SI**, that we copy here for convenience. In the FLIM measurements, we have corrected the traces by subtracting the unlabelled bead background. In ixFLIM data, the unlabelled beads are invisible.

Figure S14 Control ixFLIM and FLIM measurement on unlabeled beads. The beads are near-invisible in red detection, and display weak fluorescence in green (left image). ixFLIM data are thus background-free (middle), while FLIM excited at 525 nm and detected in 560 – 600 nm has a weak decaying background (right) that we scaled to the tail and subtracted from the FLIM TCSPC curves before analysis.

"The presence of the donor Cy3 peak in the ixFLIM spectrum is in this case immediately a qualitative proof that the FRET takes place." This is true only if there is no spectral crosstalk of the donor into the acceptor channel. The authors should phrase this comment more precisely.

We agree, that is what we referred to by 'in this case'. We have rephrased this in the revised version as "if the direct emission of the donor is negligible, the FRET-ixFLIM signal detected by acceptor fluorescence can be expressed as...". We have quantified the spectral cross-talk of the donor in the acceptor detection region to be less than 5%, by calculation from known fluorescence spectra, and by a measurement of bp20 sample with and without bleached acceptor. **We mention this on page 6 of the main text, and show the corresponding figure in the SI.** We copy the figure here below for convenience. We further note that the cross-talk can be taken into account using Eq. 7, where it is expressed by the fraction of the donor to acceptor detection efficiencies. We tested that this makes no difference in the resulting efficiencies.

Figure S13 Control ixFLIM measurement on bp20 beads (left), before and after acceptor bleaching. While the unbleached sample (red) decays with the 1.4 ns acceptor lifetime (middle), the bleached sample decays with the shorter lifetime of 0.7 ns of the donor (blue). The excitation spectra (right) show absorption of both acceptor and donor, after acceptor bleaching both signals disappear and only the donor cross-talk remains, with amplitude of 3%, which is in our analysis negligible.

"...clearly show the delayed rise of the donor peak due to the FRET." Again, a bit more precision would be helpful. The standard FRET specialist will read Figure 3e as the donor and acceptor emission signals rather than the detected excitation signals. For ixFLIM experiments, one has to think differently and the authors should be more explicit to guide the readers.

We hope that the axis label 'Excitation wavelength' together with the preceding description will prevent the reader to see time-resolved emission in Fig. 3e, now Fig. 3d. We agree that the FRET specialists might have difficulty in switching their intuition from emission to excitation, where many effects work in the opposite way, e.g., the donor signal rises instead of decaying. **We have streamlined the Figure 3 discussion, changing the order of panels and explaining in more detail why the rise of the donor signal in the acceptor channel is expected from FRET.**

Equation 6: This equation only holds if there is no detection of the signal from the donor and the fraction of excited acceptor molecules is low enough that the probability of excitation of both the donor and acceptor fluorophores on the DNA molecule is negligible. The authors should discuss this in more detail in the text.

We added these constraints explicitly into the text. Specifically, the text now states on page 6 that **"if the direct emission of the donor is negligible, the FRET-ixFLIM signal detected by acceptor fluorescence can be expressed as..."**

Concerning the simultaneous excitation of donor and acceptor, **we added in the discussion a sentence: "We note here that the excitation intensity in the linear excitation regime is needed for ixFLIM."**, as discussed above in the excited-state absorption comment 1d. In a practical sense, at intensities where there is appreciable excitation of both molecules of the same pair per laser pulse the sample will bleach rapidly.

Methods: FRET-ixFLIM description: a) The authors should describe what they mean by excitation spectra. Usually, epsilon is used for the absorption spectra. b) "In case that appreciable ratio ϕ_D/ϕ_A of the donor emission is detected as well." Why does the ratio of quantum yields play a role here, or are the authors also incorporating the detection efficiency into the quantum yield? The quantum yield is a property of the fluorophore, not of the details of the setup. Please reword.

a) We mean standard excitation spectra, **in the revised version we have used epsilon to denote them.**

b) We indeed mean the fluorescence quantum yield including the detection efficiency. **We have rewritten the expression as $\frac{\eta_D \phi_D}{\eta_A \phi_A}$,** where we separate the FL quantum yield ϕ and detection efficiency η .

*Methods: Optical Setup: a) The NA of the used objectives are missing. How sensitive is ixFLIM to the NA of the objective? b) "The intensity of excitation was attenuated such that the data collection rate was kept well below 5% of the 4 MHz repetition rate to avoid pile-up in the photon counting." 5% of 4 Mhz is well below 200 kHz. 200 kHz * 20 us pixel time ~ 4 photons per pixel maximum. Perhaps the authors should elaborate on how they can determine the interference pattern with such low statistics per time bin.*

a) **We have included the NAs of the objectives.** For apochromatic objectives, ixFLIM does not depend on the NA of the objective, it is a measurement linear in excitation intensity.

b) The original images were acquired as 256x256 pixels, and subsequently binned down to 64x64 pixels. This gives up to 64 photons per pixel per frame. In total, 301 frames are measured, and the Fourier transform uses the signal from the whole interferogram, in which the average (over delay time) maximum (over pixels) photon count is 32 photons per (binned) pixel per frame. At a single pixel, the interferogram is thus built from up to a thousand photons.

We have added Fig. S3 in the SI illustrating how the interferograms are obtained, and show an interferogram from a single bead (16 pixels). We note here that analyzing the whole ixFLIM map from a single pixel is with the given number of photons and resulting SNR unfeasible. The imaging by spectrum and lifetime can be used, and then particular regions of interest selected for the full global analysis. We copy the figure here for convenience.

Figure S3 Demonstration of the ixFLIM signal processing. Left: image of the bp20 beads binned down to 64x64 pixels. Top: TCSPC time-integrated interferogram of a single ball (4x4 pixels), the Fourier transform of which produces the raw spectrum (blue), division of that by the laser spectrum (black) leads to the ixFLIM excitation spectrum (green). Bottom: spatially integrated image produces the time-time correlation map, on x axis is the interferometric time τ , on the vertical axis the TCSPC time delay. Fourier transformation along τ produces the ixFLIM transient map.

SI: S3: spectral leak term not defined. Please define what you mean.

We mean spectral cross-talk. We have rephrased the description to this more standard terminology.

REVIEWER #2 (REMARKS TO THE AUTHOR):

This paper presents a variant of fluorescence lifetime imaging microscopy (FLIM) named ixFLIM that correlates the excitation wavelength with fluorescence decay. Rather than scanning the excitation wavelength, the authors utilize an interferometric technique. A broadband laser pulse is divided into two time-shifted, phase-stable replicas, thereby achieving spectral modulation. Data acquired as a function of the delay between pulses can be converted to a function of the wavelength via Fourier Transform. Consequently, each pixel contains the time-resolved fluorescence decay as a function of the excitation wavelength. The authors describe and analyze three experiments: Oxonol VI free and bound to BSA, FRET between Cy3 and Cy5 in DNA constructs, and HEK-293T co-transferred with nucleolar protein nucleophosmin bound to mVenus or mRFP1.

The presentation of the data contains several shortcomings. For example, residuals for the different fits are not displayed, making it impossible to ascertain the presence of systematic issues. Furthermore, uncertainties are absent from all fits throughout the text, preventing the reader from gauging the accuracy and precision of the method.

In the revised version, we have included residuals for all the fits, both one- and two-dimensional, in the main text and in the SI. Furthermore, we have calculated the uncertainties of all fitted rates and efficiencies and included these with the reported values.

Additionally, the organization of the panels within the figures is confusing, and many plots would benefit from being presented in a stacked format. Examples of such figures include 2b,g and 3c,d,f,g; and 5d,e. This is not solely an aesthetic consideration, but rather a means of presenting the data that enables the reader to evaluate it more effectively.

We have adapted figures 2,3,4 and 5 to better follow the discussion in the text, and we list a summary of the figure changes below. We have used the stacked format wherever possible with the panel order following the text flow, stacked are now panels b,d,f in Fig. 3, subpanels of Fig. 4b and 4c, and panels f,g,h of Fig. 5. We thank the reviewer for this suggestion.

List of figure changes:

Fig. 1: In panel 1a added excitation and detection region distinction, together with the sample spectra

Fig. 2: Changed the order of the panels such that it follows the text

Fig. 3: New illustration in panel 3a, order of all panels changed such that it follows their mention in the text, panels 3b,3d,3f stacked with common spectral axis, new measurement in panel 3h.

Fig. 4: Panel 4a moved here from Fig. 3 to avoid congestion. New analysis for all panels. Spectral fits in Fig. 4b and decay fits in Fig. 4c stacked with common axis.

Fig. 5: Completely new analysis of a new measurement, panels ordered to follow the text, panels 5f,5g stacked with common wavelength axis.

Fig. 6: New figure to avoid congestion of figure 5, thematically separate.

The main claim of the authors is that the extra information available in this dataset enables resolving complex structures that are unsolvable with previously published methods. While I agree with the premise that the dataset produced by ixFLIM does contain valuable information, this paper does show its potential compared to other simpler, faster, commercially out of the box, and well established FLIM

variants. For example: how does it compare with measurements performed using two or three lasers using pulsed interleaved excitation? While it contains much less information than ixFLIM, it might be enough to resolve the experimental examples described here. If this is true, what is the precision and accuracy of each one? When is it worth it to use ixFLIM? These questions must be addressed quantitatively; which could be done, for example, by slicing the ixFLIM dataset. Additionally, a spectroscopically more complex system which can only be resolved by ixFLIM must be shown.

The ixFLIM determines the full correlation between the excitation spectrum and the excited-state lifetime, no more, no less. For a system with a-priori known properties (e.g., excitation spectra and lifetimes), it will always be possible to devise a simpler experiment that uses a finite number of interleaved excitation wavelengths. In this sense, there is thus no system that can only be resolved by ixFLIM. Even if there was some, there would be no independent method to verify ixFLIM results, which is not desirable in the context of this manuscript. However, the weak point of mentioned simpler approaches is the system characterization, since spectra can shift and lifetimes change due to the local environment of the fluorophores. The situation gets even more complex, to the point of intractability, in auto-fluorescence samples with multiple emitting species.

The advantage of ixFLIM is that it allows a reliable identification of the species, measuring their spectra and lifetimes at the same time. It can thus be viewed as having 'internal references'. This is well demonstrated on the unexpectedly complex case of the FRET-DNA samples. In these samples, the donor lifetime is shortened compared to the free-donor case due to structural constraints of the bound dye, and becomes multiexponential. There is the fluorescent background of the beads themselves that affects the FLIM measurement especially for the efficient FRET case. These had to be taken into account when evaluating the FLIM measurements. Further complication is a possible formation of photo-converted species when bleaching the acceptor in the FLIM-FRET study or even during the measurement itself. On top of this, the acceptor lifetime is also dependent on the binding position along the DNA oligonucleotide. These factors are well characterized in ixFLIM, providing grounds for the detailed robust FRET quantification.

We carried out a new measurement of the interacting NPM in HEK cell nuclei, directly comparing FLIM and ixFLIM. In the new analysis, the advantage of ixFLIM is visible in that it proves the FRET from a single measurement, without the need for photobleaching the acceptor, and even quantifies the larger concentration of interacting NPM oligomers with the same FRET efficiency (and, thus, distance) in the nucleolus compared to the nucleoplasm, which is not possible by FLIM alone.

In the revised manuscript, the new analysis is presented in the NPM section. We have further included, in the Discussion on pages 12-14, sections that discuss the advantages and disadvantages of ixFLIM, its comparison to other techniques such as pulse-interleave FLIM, and its envisaged areas of usage.

The presented ixFLIM cuts at specific wavelengths (Fig. 2e,2f, Fig. 3e, Fig. 5f) are effectively slicing the ixFLIM dataset. As seen from, e.g., the donor/acceptor region traces in Fig. 3e, in case the spectra are known and distinct, couple of wavelengths carries sufficient information. **We mention this in the discussion part contrasting ixFLIM and FLIM.**

As an additional example that doesn't fit in the scope of this demonstration paper, we have applied the ixFLIM to living cells of *Vischeria* alga, with the aim of identifying possible emission of carotenoids. These have a very short excited state lifetime around 10 ps, emission in the green, and a characteristic structured absorption spectrum around 480 nm. In FLIM, one can distinguish the reddish globule in the center of the cell with 'red' emission above 560 nm and with a longer lifetime than the

surroundings. The rest of the cell, including lipid bodies, also emits in the blue/green region below (500 nm – 560 nm), and the decay is dominated by a very fast component with time constant well shorter than the 50 ps instrument response function. While this is consistent with the expected carotenoid emission, based on FLIM alone, the nature of the emitting species cannot be further identified. In ixFLIM, we observe the redder absorption of the reddish globule and bluer absorption of the rest of the cell. The ixFLIM transient map also reflects the ultrafast decay, but the excitation spectrum is resolved and can be compared to those of various carotenoids. At present, we are analysing the data and it appears that a mixture of carotenoids is present.

Also, how does it compare with FLIM using sequential scanning of the excitation wavelength, which is available in some microscopes equipped with a white light pulsed laser and an acousto-optical beam splitter?

The main difference between the interferometric scanning and sequential excitation wavelength scanning is that in the former (ixFLIM), all excitation wavelengths interact with the sample at the same time, during all the scan. This brings several advantages. The spectral and temporal resolution are independent, and there is no tradeoff between spectral resolution and excitation intensity. For samples that change in time, spectral distortions are avoided and the average spectrum over the acquisition time is acquired. Also, the interferometric ixFLIM approach inherits the general advantages of Fourier transform spectroscopy such as speed and sensitivity.

We have added these considerations in the discussion on pages 12-14, where we compare ixFLIM to existing FLIM variants and discuss its envisaged areas of use.

As a final, minor note, the statement "The measurements parameters are fully compatible with in vivo cell imaging" is way too broad. The authors assert that it takes 7 minutes to capture a dataset. A prevalent application of FLIM in cellular imaging is for monitoring the state of biosensors, molecular interactions, and post translational modifications within signaling. In many of these instances, 7

minutes encapsulates the entire process duration, and therefore this technique in its current form is too slow.

The 7 minutes that we mentioned was just an example of a relatively long measurement with high number of pixels (256x256) on the fixed cells. The measurement time depends on factors such as sample brightness, scanning speed, number of pixels, laser repetition rate, spectral range and desired spectral resolution and time resolution. On bright samples with few spatial pixels, one can easily measure the ixFLIM in tens of seconds or even faster. This is similar to the standard FLIM, which needs longer accumulation time to spatially resolve dim samples. Last but not least, the work shown in this manuscript is a proof of principle, and neither the setup nor the scan parameters were optimized for scanning speed. On the other hand, ixFLIM does need more photons than standard FLIM, and thus is not suitable to fast measurements with low photon budget such as single-molecule FRET.

We have added a discussion of the scaling of the measurement time and its impact on ixFLIM applicability on page 13 of the revised manuscript.

REVIEWER #3 (REMARKS TO THE AUTHOR):

The manuscript titled “ixFLIM: Inetreferometric Excitation”, by X et al. describes a technique for fluorescence lifetime imaging that the authors call ixFLIM. The technique allows the excitation of fluorophores at a continuous range of wavelengths, instead of using a few discrete ones which have been the best practice so far. It has the potential to separate more fluorophore species in the sample than what is possible with discrete excitation. In practice, the authors use a broadband coherent light source generated from a femtosecond Ti:sapphire oscillator, and interfere them at the sample with a specific time delay generated using an interferometric device. The variation of the delay generates different excitation profiles, and this can be transformed into 2D data which encoded both the excitation and the emission spectra of all the fluorophores present in the sample. The manuscript is well written and describes several example data which makes the use of ixFLIM clear. However, before I can recommend publishing the manuscript in Nature Comm, several points need to be addressed. These are:

1) The major concern I have is about the practicality of the technique. A multiwavelength excitation should be able to do, in practice, faster and more reliable chemical diagnostics. However, the increased instrumental complication and the time taken to record the data are apparent negative points of this method. The manuscript will benefit from a direct comparison of all the aspects between conventional FLIM and ixFLIM.

We have **significantly revised the manuscript, putting the emphasis on direct comparison of ixFLIM and FLIM** on the same samples. The reader can thus gauge the performance of ixFLIM against the standard FLIM by herself. Furthermore, **we have included** in the revised version of the manuscript, **on page 12-14 in the discussion, sections discussing the advantages and disadvantages of ixFLIM compared to other techniques such as FLIM, as well as detailed discussion of envisaged usage of ixFLIM.** As the reviewer points out, the more involved instrumentation and longer acquisition means that one needs a motivation to use ixFLIM over standard FLIM. In the revised version of the manuscript, we discuss such motivations.

2) Similarly, a comparison between the established techniques and this new one should be performed on some real or standard samples.

We have conducted additional experiments directly comparing ixFLIM and FLIM on the same samples. Specifically, **we quantitatively compare ixFLIM and FLIM** in measuring FRET in the Cy3-Cy5 DNA samples, and **we use both FLIM and ixFLIM** to prove NPM interaction in the same HEK cell nucleus. The Cy3-Cy5 DNA samples are standard FRET pairs commonly used for DNA labelling, with detailed characterization in literature **to which we compare in the revised version.** The interacting labelled NPM has also been discussed previously in literature, which we reference.

3) It is not clear how the background from the excitation which leaks into the emission channel and how it can be eliminated.

The excitation light is removed using a combination of dichroic and long-pass spectral filters. **We have added into the “Principle of ixFLIM” section a sentence “The broadband excitation region is separated from the detection region spectral region by spectral filters.” We have also included a scheme in Fig. 1b illustrating the excitation and detection regions, which we copy here for convenience.**

Figure 1 | Principle of ixFLIM. a) Standard time-domain FLIM with point scanning (x,y) and TCSPC detection (t). b) Spectral modulation of broadband excitation pulses (shaded spectra) by interference of two overlapping pulses delayed by τ . The laser spectrum (shaded regions) must cover the excitation spectrum of the sample (solid lines), and the fluorescence emission (dashed lines) should be separated by a long-pass filter. c) Fourier transform of the time resolved FLIM data leads to the transient map that correlates the fluorescence excitation spectrum (horizontal) and emission decay (vertical) at each pixel of the image. The marginals of the transient map represent the total fluorescence decay (vertical) and excitation spectrum (horizontal). Exemplary data of HEK-293T cells that are described in detail in the results.

4) Line 239: *How would this technique compare with the standard spectral deconvolution technique?*

In ixFLIM, one has the temporal additional dimension along which the spectra are resolved, reporting on the spectro-temporal correlations. Instead of the spectral deconvolution, the full global analysis can thus be used. Naturally, the ixFLIM spectrum can be integrated along the detection time, yielding the excitation spectra for which spectral deconvolution can be used. Our fits of the excitation spectra of the DNA-FRET samples by a combination of the donor and acceptor spectra, Eq. 6 and, e.g., Fig. 4b, are an example. **We have added into the discussion the following: “We note that already the time-integrated ixFLIM spectra (ixFIM) contain valuable information on the excitation, which we used, e.g., in the spectral fits for determination of FRET efficiency in the DNA samples. Established analysis of the excitation spectra such as spectral deconvolution can thus be applied to ixFIM data well. ”**

5) Line 355: *7 min seems to be a long observation time*

The 7 minutes was used as an example, the measurement time depends on factors such as sample brightness, scanning speed, number of pixels, laser repetition rate, spectral range, and desired spectral and temporal resolution. On bright samples with few spatial pixels, one can easily measure the ixFLIM in tens of seconds. This is similar to the standard FLIM, which needs longer accumulation time to spatially resolve dim samples or accumulate good enough photon statistics to characterize complex decays.

We have added a discussion of the scaling of the measurement time and its impact on ixFLIM applicability on page 13 of the revised manuscript.

Minor points:

1) Page 16: *“critical” is not typically used*

We have rephrased as simply Förster radius.

2) Fig. 3 caption: *“weighed sum” should become “weighted sum”*

We have fixed this typo.

REVIEWERS' COMMENTS

Reviewer #1 (Remarks to the Author):

The authors have done a commendable job of responding to the reviewers' comments. The added experiments and discussion improved the paper. The authors should check the English (especially in the beginning of the revised discussion section). Also, in the comparison to other methods, references are missing.

Reviewer #2 (Remarks to the Author):

In this new revised version the authors have address all my previous concerns. The added text and improved figures, allowed me and the future readers to better assess the technique. In my opinion, this work is suitable for publication.

Reviewer #3 (Remarks to the Author):

The authors have adequately addressed all my concerns about the manuscript. It may now be accepted for publication.

Reply to Reviewer Comments on Manuscript NCOMMS-23-56256A

Title: "ixFLIM: Interferometric Excitation Fluorescence Lifetime Imaging Microscopy"

Authors: Pavel Malý, Dita Strachotová, Aleš Holoubek and Petr Heřman

We are content that all three reviewers found our revision of the paper helpful and now recommend publication. We hereby wish to thank them once more for their thorough effort and help with improving the manuscript. Below we provide the answer to the remaining point of the reviewer.

REVIEWER #1 (REMARKS TO THE AUTHOR):

The authors have done a commendable job of responding to the reviewers' comments. The added experiments and discussion improved the paper. The authors should check the English (especially in the beginning of the revised discussion section). Also, in the comparison to other methods, references are missing.

We went once more through the paper for a final check of our writing.

We have included the references in the part of the Discussion where ixFLIM is compared to other methods. Keeping the length of the reference list in mind, we mostly used already cited works, and added only Ref. 63. We thank the reviewer for spotting this unintended omission.

REVIEWER #2 (REMARKS TO THE AUTHOR):

In this new revised version the authors have address all my previous concerns. The added text and improved figures, allowed me and the future readers to better assess the technique. In my opinion, this work is suitable for publication.

REVIEWER #3 (REMARKS TO THE AUTHOR):

The authors have adequately addressed all my concerns about the manuscript. It may now be accepted for publication.